# Harnessing microcomb-based parallel chaos for random number generation and optical decision making

Bitao Shen[1,8], Haowen Shu[1,8] ✉, Weiqiang Xie[2], Ruixuan Chen[1], Zhi Liu[3,4], Zhangfeng Ge[5], Xuguang Zhang[1], Yimeng Wang[1], Yunhao Zhang[1], Buwen Cheng[3,4], Shaohua Yu[1,6], Lin Chang[1,7] ✉ & Xingjun Wang [1,5,6,7] ✉

Optical chaos is vital for various applications such as private communication, encryption, anti-interference sensing, and reinforcement learning. Chaotic microcombs have emerged as promising sources for generating massive optical chaos. However, their inter-channel correlation behavior remains elusive, limiting their potential for on-chip parallel chaotic systems with high throughput. In this study, we present massively parallel chaos based on chaotic microcombs and high-nonlinearity AlGaAsOI platforms. We demonstrate the feasibility of generating parallel chaotic signals with inter-channel correlation <0.04 and a high random number generation rate of 3.84 Tbps. We further show the application of our approach by demonstrating a 15-channel integrated random bit generator with a 20 Gbps channel rate using silicon photonic chips. Additionally, we achieved a scalable decision-making accelerator for up to 256-armed bandit problems. Our work opens new possibilities for chaos-based information processing systems using integrated photonics, and potentially can revolutionize the current architecture of communication, sensing and computations.

Chaos is a fundamental phenomenon in physics that exhibits random behaviors due to its great sensitivity to small changes of conditions[1]. It has been playing key roles behind a wide range of applications for modern society: in communications, the generation of chaos guarantees the integrity of cryptographic protocols for secure networks[2–5]; in computations, the simulation of Monte Carlo problems and reinforcement learning relies on random numbers[6] generated from chaos, which is particularly essential for Artificial Intelligence (AI). Recently, chaos has also been used to enable advanced sensing technologies, such as Multiple Input Multiple Output (MIMO) radar[7,8] or Random Modulation Continuous Wave (RMCW) lidar[9–11], which are immune to

interference and thus can overcome the time/frequency congestion in ranging. Conventionally, the chaotic source used in information systems is generated from the electronic chaos (Fig. 1a1) induced by nonlinear circuits[12,13] such as application specific integrated circuit (ASIC) and field programmable gate array (FPGA)[14,15]. Despite their integration compatibility with CMOS electronics, these chaotic sources suffer from low bandwidth (on the order of hundreds of MHz[12]) and the combination of several sources is necessary for high throughput rate, which is well behind the electronic processing speed and therefore lagging the system performance. This problem gets further highlighted in nowadays parallel information architectures, where

[1]State Key Laboratory of Advanced Optical Communications System and Networks, School of Electronics, Peking University, 100871 Beijing, China. [2]Department of Electronic Engineering, Shanghai Jiao Tong University, 200240 Shanghai, China. [3]State Key Laboratory on Integrated Optoelectronics, Institute of Semiconductors, Chinese Academy of Sciences, 100083 Beijing, China. [4]Center of Materials Science and Optoelectronics Engineering, University of Chinese Academy of Sciences, 100049 Beijing, China. [5]Peking University Yangtze Delta Institute of Optoelectronics, 226010 Nantong, China. [6]Peng Cheng Laboratory, 518055 Shenzhen, China. [7]Frontiers Science Center for Nano-optoelectronics, Peking University, 100871 Beijing, China. [8]These authors contributed equally: Bitao Shen, Haowen Shu. ✉e-mail: haowenshu@pku.edu.cn; linchang@pku.edu.cn; xjwang@pku.edu.cn

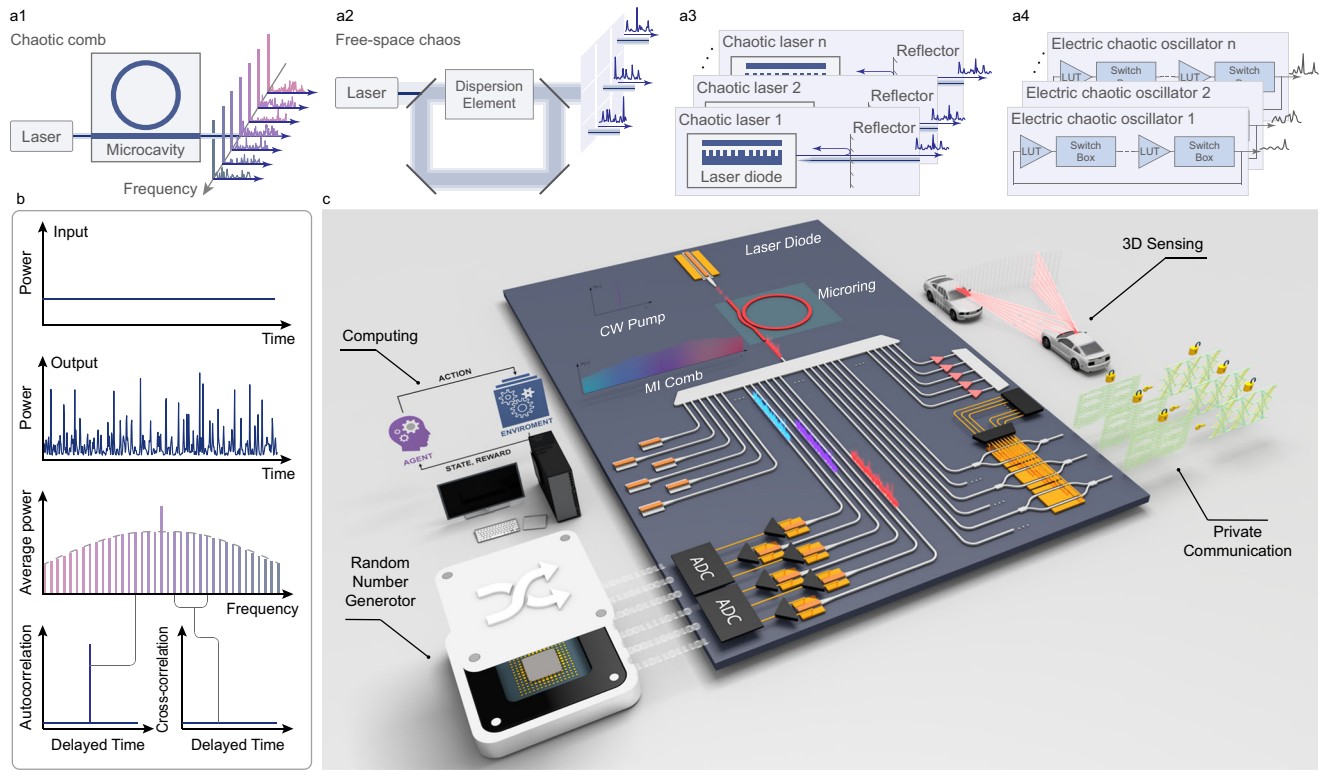

**Fig. 1 | Microcomb based massively parallel chaotic signal generation and applications. a** Different methods to obtain parallel chaotic sources, (**a1**) multiple electric chaotic oscillators; (**a2**)spatiotemporal chaos in free space; (**a3**) multiple chaotic lasers; (**a4**) chaotic comb. LUT, look-up table; Disp, dispersion element. **b** The principle of the chaotic comb function as the parallel chaotic source. As a continuous wave injected into the high-quality and high-nonlinear optical microcavity, the intracavity field will evolve into the spatiotemporal chaos. The output consists of multiple comb lines in the frequency domain. Each comb line carries a chaotic signal, whose autocorrelation function is a dirac-like function. The cross-correlation between different channels is negligible. **c** Scalable chaos-based systems empowered by chaotic combs. Using the wavelength division multiplexing technology, hundreds of chaotic sources could be distributed, detected, and processed in parallel, and employed for random number generation, reinforcement learning, lidar, radar and private communication.

multiple data channels need to be proceeded simultaneously for high system throughput.

Optical chaos[16–20] has been studied for decades to overcome the limitations of electronic approaches, by utilizing the ultra-fast nature of optical processes (Figs. 1a, 2–3). Chaotic lasers[21–23] have been widely used for this purpose, by perturbing the laser cavities with external injection or optical feedback[24], which leads to highly nonlinear dynamics and chaotic behavior. Chaotic lasers have achieved up to 50 GHz[25] chaotic bandwidth, surpassing electronic approaches. However, the integration and parallelization of chaotic lasers have encountered remarkable challenges. Multi-wavelength lasers often exhibit high correlation between different wavelengths due to the competition between longitudinal modes[26], making chaotic lasers single-channel[27] devices. Previously, parallelization relied on incompatible spatial multiplexing[16,28] or laser arrays[29,30]. Moreover, the requirement for strict injection or feedback configurations complicates miniaturization and widespread usage. These limitations hinder the practical deployment of optical chaos[30].

The use of a chaotic comb (Figs. 1a, 4 as a parallel chaotic source offers a promising alternative, as spatiotemporal chaos can be achieved by pumping an optical nonlinear microcavity with sufficient power. In frequency domain, the comb is consisted of well-separated comb lines, each exhibiting poor coherence[31–33]. This promises tremendous parallelism provided by optics through wavelength-division-multiplexing technology. Recently, chaotic combs have been employed as parallel optical sources for optical coherence tomography[34,35] and parallel ranging[11]. Especially for parallel ranging, the chaotic properties of chaotic combs were harnessed for unambiguous and interference-free Lidar with simplified systems[11,36].

However, the parallelization in previous works was based on frequency domain rather than on the chaotic signal carried by each channel. For parallel chaotic sources, channels should be uncorrelated with each other, which is vital to applications ranging from private communication[4,37] and key distribution[5] to chaotic lidar/radar systems[7,36]. However, it is still unclear whether chaotic combs can function as parallel chaotic sources.

In this work, we fill this gap by providing a full investigation of chaotic comb lines, and introduce the optical chaotic comb as a massively parallel chaotic source with low inter-channel correlation as shown in Fig. 1b. For the first time, to the best of our knowledge, we verified inter-channel orthogonality relation between each tooth of a chaotic microcomb in both theory and experiment, which demonstrated its parallelization capability. By using an advanced heterogeneous photonic platform, aluminium gallium arsenide on insulator (AlGaAsOI) on silicon substrate[38,39], we demonstrate a massively parallelled chaotic source with hundreds of wavelength channels on chip, that is compatible with current photonic foundry production. Based on this capability, we explore two advanced applications. A 15-channel random number generation system with 20 Gbps bit channel rate is achieved using silicon photonic WDM and receivers, whose aggregate rate is one order of magnitude higher than previous on-chip random number generators. Detecting two chaotic microcombs with commercial photodetectors, the aggregation rate can reach 3.84 Tbps, with 32 channels and 120 Gbps per channel rate. This is the highest generation rate for optical-chaos-based systems. We also show the unprecedented advantages of this strategy in optical computations by accelerating the decision making of multi-armed bandit problem. Our work paves the way for information processing on chip

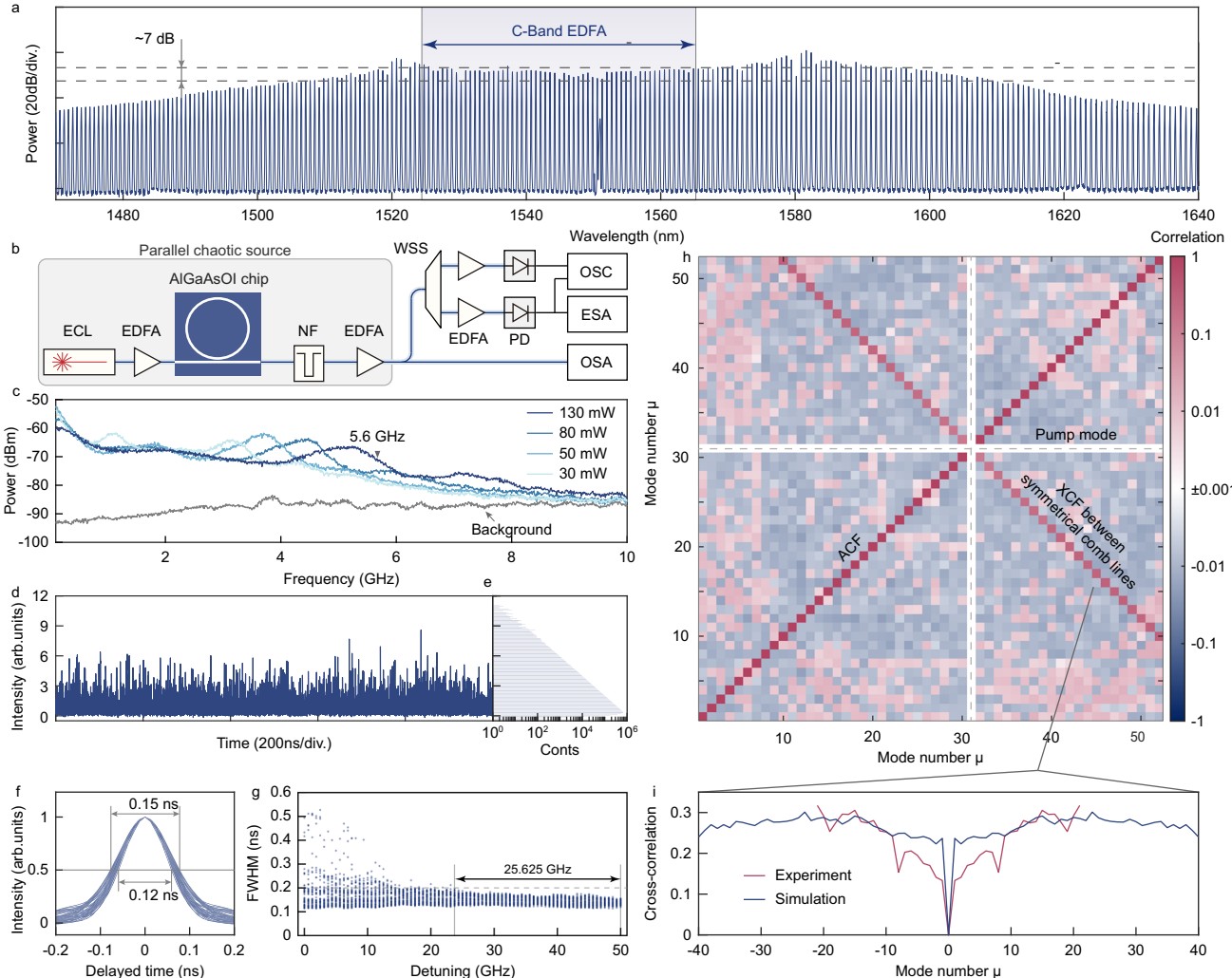

**Fig. 2 | The characterization of chaotic combs. a** The optical spectrum of the generated chaotic comb. **b** Setup for characterizing the chaotic comb. ECL external cavity diode laser, EDFA erbium-doped fiber amplifier, NF notch filter, WSS wavelength selective switch, PD photodetector, OSC oscilloscope, ESA electrical spectrum analyzer, OSA optical spectrum analyzer. **c** Radio frequency noise spectrum of chaotic combs pumped by different power levels. The gray line shows the spectra without optical input. **d** The time serial of a single comb line recorded by the oscilloscope. **e** The amplitude distribution of the time serial shown in (**d**). **f** The autocorrelation function (ACF) for all comb lines in C band. **g** The full width at half maximum (FWHM) of the ACF for different comb lines varies with the detuning of the pump laser. **h** The correlation between different comb lines. **i** The cross-correlation between symmetric comb lines in experiment (red line) and simulation (blue line).

## Results

### Properties of massively parallel chaotic combs

Firstly, we verify the ability of microcombs in chaotic states to act as massively parallel chaotic sources. Chaotic bandwidth is one of the key metrics of chaos and Fig. 2c shows the radio frequency spectra of chaotic combs under different pump powers. Due to the high non-linear effect, a chaotic comb with ~GHz bandwidth is reachable with tens of mW pump power. The bandwidth of the chaotic signal broadens with a stronger pump. As the on-chip pump power reaches 130 mW, the chaotic comb shows a broad radio frequency noise with a 10 dB-bandwidth of up to 5.6 GHz. The bandwidth is currently limited by the free-carrier-absorption (FCA) and the three-photon-absorption (3PA) in the AlGaAsOI waveguide, as discussed in the Supplementary Note 8. Nonetheless, the bandwidth is already compatible with that of the optical-feedback chaotic laser. In addition, the broad chaos, starting from the very low-frequency baseband, holds consecutively

using integrated, high performance chaotic sources, which will lead to many new opportunities in private communications, computation and ranging for integrated photonics (Fig. 1c).

strong components within its noise band, which is power-efficient for radio applications considering a low-pass response in the optical receiver for RF signal generation[7].

To characterize the intra- and inter-channel chaotic properties (see Supplementary Note 1), each comb line is filtered out from a chaotic microcomb with 130 mW pump power (Fig. 2a). The comb is rendered as the trapezoidal shape in log scale, with an approximate power variation of 7 dB within the C band. Before each filtered comb line is sent into the photodetector and afterward recorded, the pump mode is suppressed and the remaining signals are amplified together, as shown in Fig. 2b. The time domain signal of a comb line is depicted in Fig. 2d. Due to the intracavity field undergoing the spatiotemporal chaos, the amplitude of the comb line changes rapidly and intensely, totally different from the case of the localized comb state. The extreme events are also captured in the amplitude distribution (Fig. 2e), where a long tail exists at high intensity. The quality of the chaotic signal is valued by the autocorrelation function (ACF) in Fig. 2f. The ACFs of the 51 comb lines possess a Dirac-like shape. Without the need for a feedback loop to estimate a chaotic state as the chaotic laser, no time delayed signature is observed in the ACF of chaotic combs. It is also

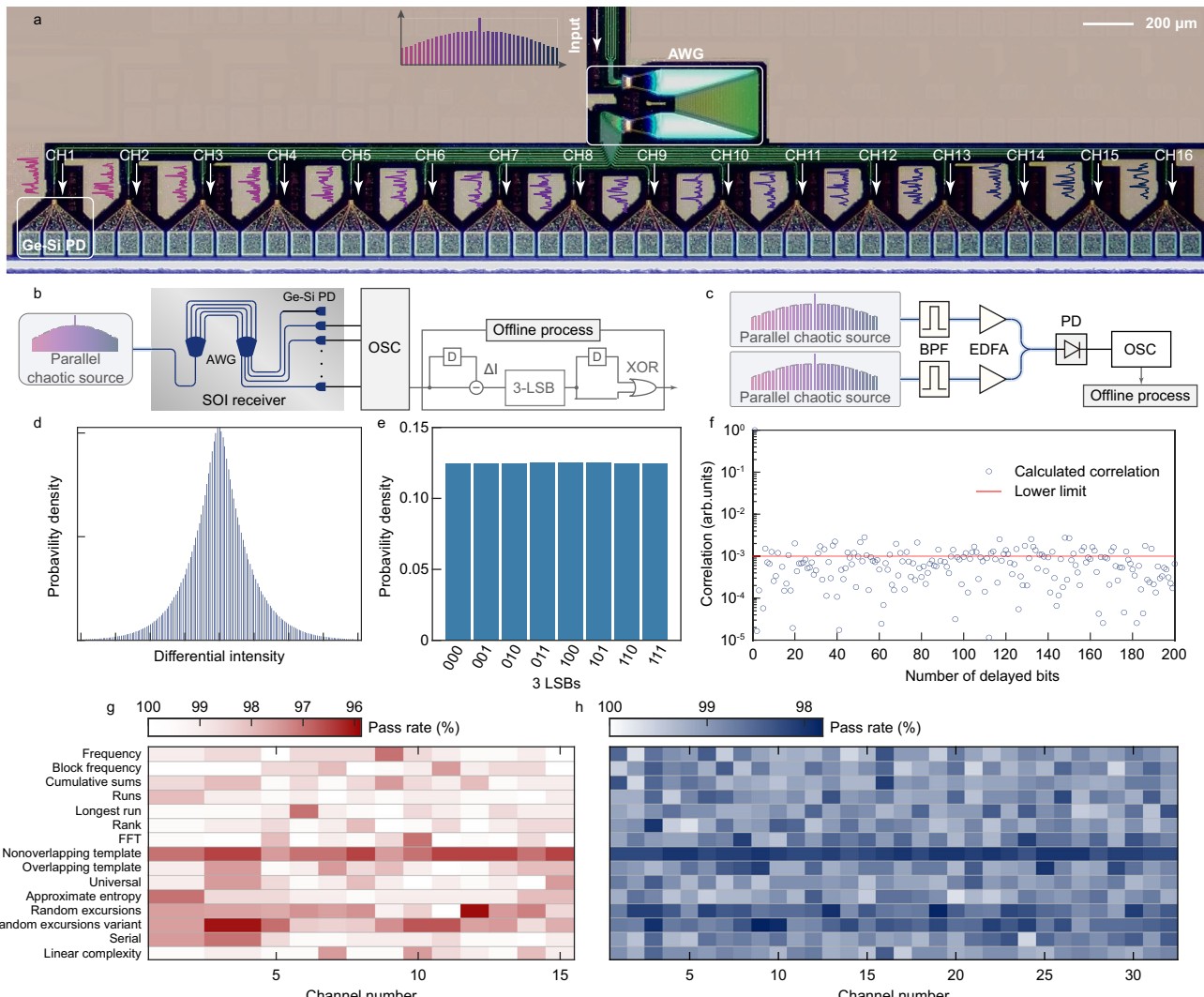

**Fig. 3 | The parallel random bit generator based on a microresonator and a SOI chip. a** Optical microscope photograph of the SiPh receiver. **b** The set-up scheme for parallel random number generation; AWG arrayed waveguide grating, PD photodetector, OSC oscillator, D delay unit, XOR exclusive-OR, BPF band-pass filter, EDFA erbium-doped fiber amplifier. **c** The setup for random bit generation using two chaotic combs. **d** The possibility distribution function of the differential data. **e** The distribution of the extracted 3 LSBs. **f** The ACF of the generated bit sequence. The red line indicates the lower limit determined by $1/\sqrt{n}$. The NIST SP800-22 test results for signals detected by SOI PD with setup shown in (**b**) (**g**) and commercial InP PD with setup shown in (**c**) (**h**).

worth noting that the strong Kerr effect in an AlGaAsOI micro-resonator, which holds the highest third-order nonlinear coefficient among all integrated nonlinear platforms[36,38], helps to achieve the full width at half maximum (FWHM) of all comb lines smaller than 0.15 ns. Figure 2g shows the FWHM of the ACF changes with the detuning. Due to the self-locking induced by the intracavity thermal effect[39,40], the FWHM for all comb lines can be maintained within 0.2 ns for a detuning range larger than 25 GHz, indicating remarkable stability and inter-channel consistency.

Despite the simultaneous generation of multi high-quality chaotic signals demonstrated above, the chaotic comb cannot be employed as a parallel chaotic source unless channels are orthogonal to each other. The orthogonality is quantified by the correlation between channels. In our experiment, we examine the inter-channel correlation by filtering and detecting every two comb lines simultaneously. The correlation of each pair of comb lines in the C band is shown in Fig. 2h. Because the pump mode is suppressed, the correlation between the pump mode and others is not presented here. Two significant lines with relatively strong correlations can be observed. One is the main diagonal, which is autocorrelation. The other is

symmetric with respect to the pump mode, indicating an obvious correlation between symmetrical comb lines. The correlation between symmetrical comb lines is also captured in our simulation based on the Lugiato-Lefever equation (LLE), shown in Fig. 2i and Supplementary Note 4. As the energy to stimulate comb lines is originated from the intracavity pump laser, the symmetrical comb lines should be naturally correlated with each other due to the four-wave-mixing process. Except for the symmetrical modes, the correlation between comb lines is lower than 0.04, which is low enough for applications requiring parallel chaotic sources[7,36,37,41].

Hundreds of channels, the uniform power distribution, the negligible cross-correlation between different channels, and the comparable chaotic bandwidth with chaotic lasers, all these properties suggest the chaotic comb is an attractive massively parallel chaotic source. In addition, different from its localized partner, the soliton comb, the chaotic comb can be generated in a simple and distinct manner, by slowly sweeping into a high-quality optical nonlinear microcavity without the blocking of the intracavity thermal effect. More attractively, the intracavity thermal effect can facilitate the stabilization of the chaotic state[39,40].

## Integrated parallel random bit generation

As proof of the chaotic comb functioning as a parallel chaotic source, we combine the microcomb and a silicon photonic (SiPh) integrated circuit to realize massively parallel random bit generation (RBG). Figure 3b shows the detection scheme using the SiPh receiver. Before injection into the SiPh chip, the chaotic comb output from the microring chip passes a FBG to suppress the pump mode and is amplified by a commercial DWDM EDFA. To compensate for the insertion loss of the SiPh chip, one selected comb line is filtered and amplified before injection into the SiPh chip. The SiPh chip is constituted of a 16-channel arrayed wavelength grating (AWG) and arrayed Ge-Si photodetectors. The average channel spacing of the AWG is about 180 GHz, fitted with the 2 FSR of the chaotic comb. The selected comb line is selectively coupled into one PD passing the AWG. The signal detected by the Ge-Si PD is sent to the oscilloscope, recorded at the sampling rate of 20 GSa/s.

Ensuring the relatively low correlation between adjacent sampled data, the recorded data is down-sampled to 6.67 GSa/s. As shown in Fig. 2e, the intensity distribution of the output $I$ shows an asymmetric shape. To symmetrize the distribution, a delay difference is employed to the raw data, obtaining $\Delta I$ with a symmetric distribution as shown in Fig. 3d. The symmetric distribution is favorable for unbiased bit extraction[42]. The differential data $\Delta I$ is digitalized to 8 bits and the 3 least significant bits (LSB) are used for random bit generation. A normal distribution of the extracted 3 LSBs (see Fig. 3e) indicates that the generated bit sequence contains different bit patterns with equal frequencies. Then a self-delayed exclusive-OR (XOR) process is employed to remove the residual bias as shown in Fig. 3b. Figure 3f shows the ACF of the generated bit sequence and the correlation is around the lower limit $1/\sqrt{n}$, where $n = 10^6$ is the length of bit sequence. A strict test of the generated random bits is carried out by the NIST SP 800-22, a standard statistical test suite. As shown in Fig. 3g, all of the 15 channels obtain pass rates >96.5%, within the acceptance range for the NIST SP 800-22 test. The generated random bits also pass the Diehard test successfully, shown in Supplementary Note 11.

The generation rate of single chaotic comb is mainly limited by the chaotic bandwidth, which is degenerated by the nonlinear absorption. To compensate this, here we provide a method by a dual-comb scheme. The schematic is shown is Fig. 3c, where two combs are pumped by two lasers with a frequency difference around 4 GHz. In experiment, two microcavities are packaged with temperature controllers and the chaotic combs are generated by sweeping the resonant frequency by tuning the chip temperature, while the pump lasers are kept fixed. The comb teeth of the two chaotic combs are filtered, amplified and combined, before being sent to the photodiode. The time domain waveforms of each tooth pairs are recorded at a sampling rate of 80 GSa/s. The raw data are down-sampling to 40 GSa/s and processed as described above. In this case, the single channel generation rate can be increased to 120 Gbps, which is comparable to the random bit generator based on chaotic lasers. 32 channels on the same side of the pump mode are recorded, processed, and successfully pass the NIST SP 800-22 test (see in Fig. 3h), corresponding to an aggregation rate of 3.84 Tbps.

In Table 1, we present a comparison of different random bit generation schemes. Due to the massively parallelism provided by chaotic combs, the total bit rate of chaotic-comb-based RBG shows the highest generation rate among optical-chaos-based methods. As the lack of data in published articles, the power consumption is not listed in Table 1. Considering simple setup with one pump laser and one microcavity, the generation of optical chaotic sources based on chaotic combs is power-efficient and low-cost compared with chaotic lasers[43,44], which require at least one laser diode per channel. In addition, our work gives the demonstration of parallel chaotic signal detection employing the integrated SiPh chip. Although the generation rate of chaotic-comb-based RBG is lower than ASE (amplified spontaneous emission)-based RBG with spectral or spatial parallelism, the use of chaotic systems offers the ability to synchronize the output of two chaos generators[4,37], which has been well studied for chaotic lasers[2,5] but not demonstrated yet for chaotic combs. The synchronized systems can be arranged at the transmitter and receiver sides respectively. This synchronization enables key distribution and private communication, which is not possible with stochastic sources such as ASE and random lasers.

## Computation acceleration based on chaotic combs

Optical chaos, with its fast and complex internal time evolution, is a powerful entropy source that can be used for exploration purposes[45]. Chaotic lasers have been successfully employed to solve the multi-armed bandit problem (MAB)[29,46,47], a fundamental problem of reinforcement learning. To scale up the problem exponentially, a parallel scheme is required that employs parallel chaotic signals or is based on time-division-multiplexing[48]. In this section, we propose the use of the

**Table 1 | Comparison of different schemes of random bit generation based on optics**

| Scheme | Total bit rate (Gbps) | Single channel rate (Gbps) | Channel number | Bandwidth (GHz) | Integrated source | Integrated processor |
|---|---|---|---|---|---|---|
| Random laser[64] | 540 | 540 | 1 | 29[b] | N | N |
| Quantum phase fluctuation of laser[65] | 117 | 117 | 1 | ~ | N | N |
| Bi-phase state of OPO[66] | 0.002 | 0.002 | 1 | ~ | Y | N |
| Vacuum state Fluctuation[67] | 18.8 | 18.8 | 1 | ~2.3[b] | Y | Y |
| ASE of EDFA[68] | 16800 | 400 | 42 | <40[a] | N | N |
| Laser diode (ASE)[41] | 254000 | 2000 | 127 | 315[a] | Y | N |
| Chaotic laser[69] | 320 | 320 | 1 | 16.7[a] | N | N |
| Chaotic laser[70] | 1200 | 600 | 2 | 26[a] | N | N |
| Chaotic laser[71] | 2240 | 320 | 7(3)[c] | ~ | N | N |
| Chaotic laser[72] | 10 | 10 | 1 | 11.6[b] | Y | N |
| Chaotic laser[73] | 21.1 | 21.1 | 1 | 7[b] | Y | N |
| Chaotic comb | 300 | 20 | 15 | 5.6[b] | Y | Y |
| (this work) | 3840 | 120 | 32 | 9.6[b] | Y | N |

[a]Effective bandwidth, defined as the width of spectral segment that accounts for 80% of the total power.
[b]10 dB bandwidth.
[c]7 channels of random number generation, by linear combining three channels of optical chaotic signals.

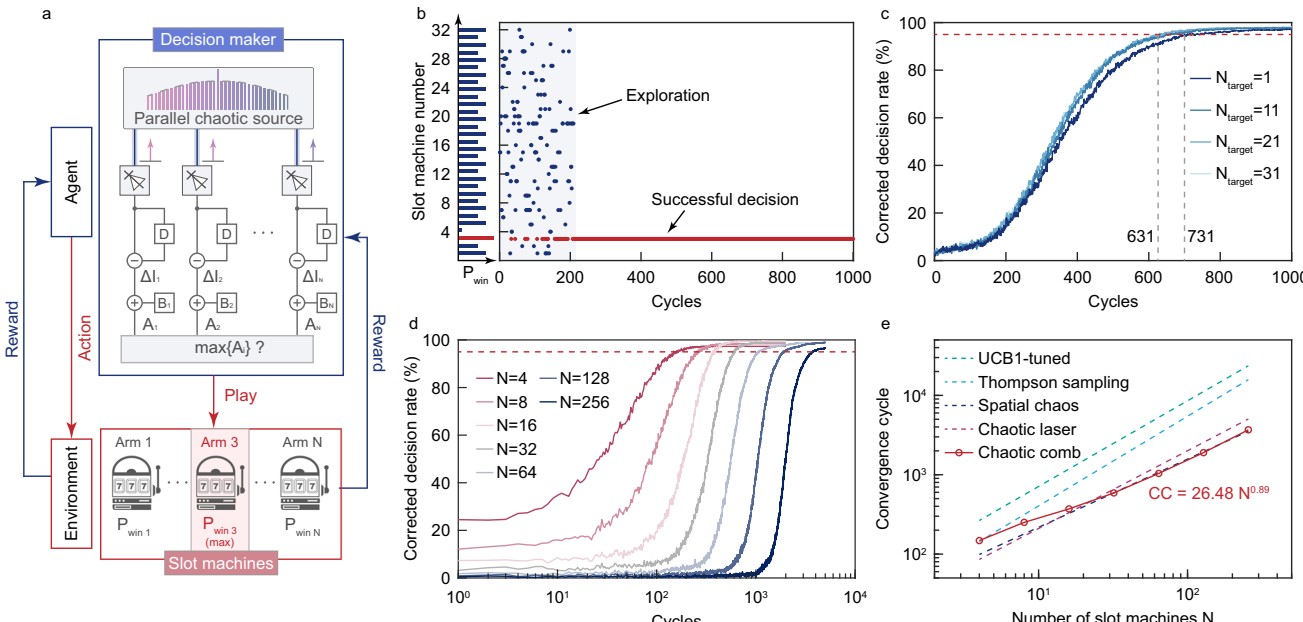

**Fig. 4 | Multi-armed bandit problem solving based on chaotic combs. a** The scheme of the optical decision making based on the parallel chaotic source. **b** One decision process for 32-armed bandit problem. The left figure shows the hit probability distribution of 32 slots, where the third slot have the highest hit probability 0.9. **c** The evolution of corrected decision rate with the increase of cycles. The red dashed line marks the corrected decision rate of 95%. **d** The evolution of corrected decision rate under different scales. **e** The comparison of scalability between chaotic-comb-based decision maker and other methods.

chaotic comb as a massively parallel chaotic source to solve the MAB. Figure 4a shows the schematic of the optical decision making. There are $N$ slot machines with a reward possibility $p_i(i = 0 \sim N-1)$ respectively. The decision maker needs to find out the slot machine with the highest reward possibility by consecutively playing the slot machines. For each play, the decision maker chooses and plays one slot machine. The decision maker will change its selection strategy based on whether a reward is obtained. In the decision maker, $N$ chaotic signals $\Delta I_i(t)$ ($i = 0, 1, 2, 3$) are detected and add with bias values $B_i(t)$ respectively to produce $A_i(t)$. Each channel is correlated with one slot machine. For each play, the slot machine with the highest biased signal $A_i(t)$ is selected and played. As continuously playing the game, bias values will be tuned based on tug-of-war method[47] (see Methods).

In experiment, we detect 32 channels at the short-wavelength side of the pump mode one by one, to solve the 32-armed bandit problem. The hit probabilities of the 32 slots are set as follows: $P_1 = 0.7$, $P_2 = 0.5$, $P_3 = 0.9$, $P_4 = 0.1, \ldots, P_{2j-1} = 0.7$ and $P_{2j} = 0.5$. As shown in Fig. 4b, the chaos-based decision maker would initially explore a wide range to identify the slot machine with the highest hit probability. After sufficient exploration, the slot machine 3 is identified as the best one and frequently selected, indicating successful decision making. To evaluate the performance of the decision making based on chaotic combs, the slot machines are played $C = 1000$ cycles consecutively, and this process is repeated $T = 1000$ times. For $t$-cycle, the correct decision ratio (CDR) is defined as $T_{hit}/T$, where $T_{hit}$ represents the number of times a reward is obtained at the $t$-th cycle among the $T$ processes. A CDR above 95% is viewed as a successful decision and the first cycle getting a CDR > 95% is defined as the convergence cycle (CC). Figure 4c shows the CDR evolution along with the increase of cycles. Under the same system condition, the CDR evolution remains consistent across different $N_{target}$, which represent the number of the slot with the highest hit probability.

The scalability is a critical characteristic of the decision maker, where the convergence cycle will exponentially increase as the rise of problem scales. To assess the ability of solving large scale problems, we use the chaotic comb to solve the MBA problems with the number of slot machines $N = 4, 8, 16, 32, 64, 128, 256$. Limited by the gain bandwidth and the spectrum width of the chaotic comb, the number of

available channels is 32. For $N > 32$, we record the 32 channels repeatedly to obtain enough channels of chaotic signals. As shown in Fig. 4d, the chaotic-comb-based decision marker can solve the MBA problems even with a large scale. Figure 4e gives the performance comparison between chaotic-comb-based decision marker and other methods[47]. Fitted with the power function, the relationship between the convergence cycle $CC$ and the number of slot machines $N$ can be described as $CC = 26.48 \cdot N^{0.89}$. Compared with widely-used algorithm such as UCB1-tuned (upper confidence bound 1-tuned) algorithm and Thompson sampling algorithm, methods based on optical-chaos show smaller scaling exponents, suitable for large-scale problems. Among the optical chaos, the decision maker shows a superb convergence speed, close to the spatial-chaos based method. While compared with the spatial chaos system, the chaotic microcomb benefits from integrated optoelectronics, providing a compact, low-cost, mass-producible decision marker. Despite having fewer channels compared to the spatial chaos system[47], the chaotic comb feathers high generation rate (40 GSa/s employed here for decision making) and the adoption of time-multiplexing[48] can be a promising approach to scale up the system, even though it may result in a tradeoff between speed and scalability.

## Discussion

The bandwidth and the parallelization capability demonstrated here can be further improved by optimizing the design as well as fabrication for the microresonator. Currently the microresonator in this work is critically coupled, and by using an over-coupled structure instead, higher conversion efficiency, wider optical spectrum and higher power per comb line could be obtained without degenerating the chaotic properties. Another factor which limits the chaotic bandwidth currently is the intra-cavity nonlinear loss[49,50] (discussed in Supplementary Note 8), which mainly comes from the three photon absorption and consequently free carrier absorption, leading to relatively high side lobes. This can be solved by employing an integrated PIN structure for the waveguide or working at a longer-wavelength band[51]. Combining these strategies, combs with chaotic bandwidth above 10 GHz can be expected with 100 mW pump powers. Furthermore, the spectral coverage of the comb can be further extended through dispersion

engineering, accessing much more chaotic comb channels than those at C band in this work. By combining all these strategies, a parallel random number generator with beyond 3 Tbps total rate can be achieved, by using only one comb source (detailed discussions can be found in Supplementary Note 6 and 10), which is even better than the best benchtop chaotic system.

One key advantage of our approach is the scalability. The AlGaAsOI platform is fabricated by heterogeneous integration on Si substrate[52], and thus is compatible with the most widely used platform of silicon photonics[53–55]. Such integration has recently been demonstrated[56], suggesting that our chaotic microcomb can be seamlessly implemented in fully integrated photonic systems, together with diverse silicon photonic engines such as optical I/O, optical computation[57,58] unit or optical phase array for sensing[59]. Other materials, such as silicon nitride[60], hydex[61,62] or lithium niobite[63], can also be used for realizing chaotic microcombs, and potentially can also be integrated with silicon photonics.

Besides the applications we showed in this work, the parallel chaotic sources can also benefit many other applications. It can be used to generate chaotic signals for MIMO radar[7,8]/LiDAR[9–11], which will lead to higher resolution due to the large bandwidth and higher energy efficiency. One great advantage of the chaotic microcomb, compared with previous widely used electronic chaotic source, is the capability of directly generating optical signals. Its output thus can be transmitted over long distance for communications, or emitted directly to free-space for sensing. As a result, this integrated, massively parallel chaotic source holds the promise to rewrite the paradigm of information technologies in the future.

## Methods

### Design and fabrication of the devices
The ring waveguides of the AlGaAsOI resonator were designed to work in anomalous dispersion with a cross-section of 400 nm × 650 nm. The width of the bus waveguide at the facet was designed to be 200 nm for efficient chip-to-fiber coupling. The fabrication of AlGaAs microresonators was based on heterogeneous wafer bonding technology. The epitaxial wafer growth was accomplished using molecular-beam epitaxy (MBE). A 248 nm deep-ultraviolet (DUV) stepper was used for the lithography. A photoresist reflow process and an optimized dry etch process were applied in waveguide patterning to minimize waveguide scattering loss. More fabrication details can be found in[38] and[52]. The silicon photonics PIC was fabricated on a 200 mm SOI wafer with a silicon-layer thickness of 220 µm and a BOX layer thickness of 2 µm using CMOS-compatible processes at CompoundTek Pte. The PD exhibits 3 dB bandwidths more than 20 GHz. In our experiment, lensed fibers with different mode field diameter (MFD) were selected for the AlGaAsOI and SOI chips; the coupling loss is 3–5 dB per facet for AlGaAsOI waveguides and 2–3 dB per facet for Si waveguides.

### Characterization of chaotic combs
The pump laser is provided by an external-cavity-diode laser (Toptica CTL 1550). For the dispersion estimation, the pump laser is scanning from 1515 nm to 1630 nm. The pump power is attenuated by a tunable optical attenuator to avoid the resonance distortion caused by the thermal drift. The resonant frequencies are recorded for dispersion calculation by $D_{int}(\mu) = \omega_\mu - \omega_0 - \mu D_1 \approx \frac{D_2}{2}\mu^2$, where $\omega_\mu$ is frequency of the $\mu$–th resonance and $\mu = 0$ is the central mode. $D_1/2\pi$ is the free spectral range (FSR) and $D_2$ is the second-order dispersion. A positive $D_2$ indicates the anomalous dispersion shown in the Supplementary Note 6.

For chaotic comb generation, the pump laser is boosted to 26 dBm by a high-power EDFA and the ASE is partially filtered by a tunable band-pass filter. Lensed fibers are employed to couple the pump laser into the microresonator chip and collect the output chaotic comb into the test link. The remaining pump light is suppressed by a fiber bragg grating and amplified by a DWDM EDFA. Each comb line in the C band is filtered and the chaotic signal is recorded by an electrical spectrum analyzer in the frequency domain and the oscilloscope in the time domain. For the data $I$ recorded by the oscilloscope, the autocorrelation function $ACF$ and the cross-correlation function $XCF$

$$ACF_i(\tau) = \frac{\langle \delta I_i(t+\tau) \cdot \delta I_i(t) \rangle_t}{\langle \delta I_i^2(t) \rangle_t} \quad (1)$$

$$XCF_{ij}(\tau) = \frac{\langle \delta I_i(t+\tau) \cdot \delta I_j(t) \rangle_t}{\sqrt{\langle \delta I_i^2(t) \rangle_t \cdot \langle \delta I_j^2(t) \rangle_t}} \quad (2)$$

are calculated. $\delta I_i(t)$ is the fluctuation of the recorded data $I_i(t)$ from channel $i$ and is equal to $I_i(t) - \langle I_i(t) \rangle_t$. $ACF_i(\tau)$ indicates the autocorrelation function of channel $i$ and $XCF_{ij}(\tau)$ represents the cross-correlation function between channel $i$ and $j$. The maximum of $XCF_{ij}(\tau)$ is used to valuate the correlation between channel $i$ and $j$, as shown in Fig. 3h.

### Random bit generation
In the experiment of random bit generation based on the SiPh chip[55], the microring with FSR ~90 GHz is used to fit the average channel spacing of the integrated AWG. With 26 dBm pump laser, a chaotic comb is generated and individual comb lines are coupled into the SiPh chip and selectively injected into certain Ge-Si PDs, after being filtered by a notch filter and amplified. The input power to the SiPh chip is maintained around 14 dBm to compensate the insertion loss of the edge coupler (~2 dB per facet) and the AWG (5–8 dB). The bandwidth of the Ge-Si PDs is about 20 GHz with a bias voltage of −3 V, enough for the detection of the chaotic comb. For each channel, $2 \times 108$ points are recorded at the sampling rate of 20 GSa/s. For the random bit extraction, the raw data detected by the Ge-Si PD are down-sampled to 6.67 GSa/s for a low correlation between continuous sampled data. Then, a delay difference $(I(i+6) - I(i))$ is employed to the raw data $I$, obtaining $\Delta I$ with a symmetric distribution. The differential data is then digitalized into an 8 bit binary number. After that, we discard the most significant bits (MSBs) and keep 3 bits of LSBs (M-LSBs) as the final output of RBG. The generation rate is thus equal to 3 times the sampling rate for each channel, where the random bit generation of 20 Gbps per channel is demonstrated for the SiPh chip. For the NIST SP 800-22 test, 200 and 1000 bit sequences are employed as the input of the test respectively for Fig. 3g and Fig. 3h, where each bit sequence contains $10^6$ bits.

### Optical decision making
The optical decision making process presented here is based on[47]. In our experiment, 32 channels are filtered and recorded one by one. The recorded data are employed for an off-line decision-making process. At cycle $t$, the slot machine with the highest biased value $A_i$ is selected and played. The biased value $A_i$ is calculated based on

$$A_i(t) = \Delta I_i(t) + kB_i(t) \quad (3)$$

$B_i$ is the bias, which is tuned based on the estimated hit probabilities $\hat{P}_i$, given by:

$$B_i(t) = Q_i(t) - \frac{1}{N-1}\sum_{i' \neq i}^{N} Q_{i'}(t)$$

$$Q_i(t) = \Delta W_i - \left( \hat{P}_{top1} + \hat{P}_{top2} \right) L_i \quad (4)$$

$$\Delta = 2 - (\hat{P}_{top1} + \hat{P}_{top2})$$

$$\hat{P}_i = \frac{W_i}{T_i}$$

Where $T_i$ is the time that the slot machine $i$ is selected, $W_i(L_i)$ is the time of win (loss) as selecting slot machine $i$. $\hat{P}_{top1}$ and $\hat{P}_{top2}$ are the highest and second highest estimated hit probabilities. All parameters above are determined by the game process, except for $k$. The value of $k$ is tuned to obtain the balance between exploration and convergence. The decision process will converge fast under a large $k$, where a large bias is obtained after a few cycles, and vice versa. For $N = 4, 8, 16, 32, 64, 128, 256$, $k$ is set to 0.11, 0.17, 0.23, 0.33, 0.48, 0.58, 0.67 respectively, determined by sweep the $k$ value

under different $N$ as illustrated in Supplement note 13.

## Numerical simulation

To get a deep insight into the process of the chaotic comb evolution, numerical simulations based on the Lugiato-Lefever equation

$$t_R \frac{\partial E(t,\tau)}{\partial t} = \left[ -\left(\frac{\alpha}{2} - i\delta_0\right) + iL\frac{\beta_2}{2}\frac{\partial^2}{\partial\tau^2} \right] E + iL\gamma|E|^2 E + \sqrt{\theta}E_{in} \quad (5)$$

are carried out under different conditions. $E$ stands for the intracavity temporal fields and $\alpha$ is the roundtrip cavity loss factor. $\delta_O$ is the detuning between cold-cavity resonant frequency and pump laser. $t_R$ is the roundtrip time of the primary mode and $L$ is the roundtrip length. The pump filed is coupled into the cavity by $\sqrt{\theta}E_{in}$, where $\theta$ is the waveguide coupling coefficient and $E_{in}$ is the pump field. $\beta_2$ represents the second-order dispersion coefficients. More simulation result about the chaotic comb can be found in Supplementary Note.

## Data availability

The data that supports the plots within this paper and other findings of this study are available on Zenodo (https://doi.org/10.5281/zenodo.8105301). All other data used in this study are available from the corresponding authors upon reasonable request.

## Code availability

The codes that support the findings of this study are available from the corresponding authors upon reasonable request.

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

## Acknowledgements

The authors thank Fenghe Yang, Yan Zhou in Peking University Yangtze Delta Institute of Optoelectronics for microcomb packaging support. This work was supported by the National Key Research and Development Program of China (2021YFB2800400, X.W.), National Natural Science Foundation of China (62235002, 12204021, X.W.), Beijing Municipal Science and Technology Commission (Z221100006722003, X.W.), Beijing Municipal Natural Science Foundation (Z210004, X.W.), Nantong Science and Technology Bureau (JC2021002, Z.G.).

## Author contributions

The experiments were conceived by B.S. and H.S. The devices were designed by H.S., L.C., Z.L., and B.C. The AlGaAsOI microresonators were fabricated by W.X. and L.C. The chip was packaged by Z.G. The microcomb simulation and modelling were conducted by B.S. The experiments were performed by B.S. and H.S., with the assistance by R.C. X.Z., Y.W. and Y.Z. The results were analyzed by B.S. All authors participated in writing the paper. The project was under the supervision of H.S., S.Y., L.C., and X.W.

## Competing interests

The authors declare no competing interests.
