## [Peer Review File · Nature Communications]

REVIEWER COMMENTS

Reviewer #1 (Remarks to the Author):

Please, see the attachment.

Reviewer #2 (Remarks to the Author):

In this manuscript, the authors report a demonstration of a chaotic generator that can provide up to 15 channel random bits at high speed, based on a compact AlGaAsOI integrated photonic nonlinear microresonator along with a silicon photodetector array. These chip-based chaotic sources were applied to potentially important applications such as parallel random bit generation and computation acceleration, supported with concrete and comprehensive experimental demonstrations. Overall, I find the results novel and believable, and the manuscript is well-written. Although the chaotic optical combs in microresonators have been reported (the authors have cited prior work), the large-scale parallelization of the random bit generation and its application in bandit problem-solving using integrated photonic devices will be of interest to a broad audience. For this reason, the presented results warrant the publication of this work provided addressing the following comments:

1. To clearly position the breakthrough claimed in this work, a table that summarizes the state-of-the-art chaotic sources, in particular for those using photonics, is suggested to be provided in the main manuscript.
2. Despite the chaotic state studied in this work, there are various chaotic comb states such as the chaotic soliton state under anomalous dispersion and the chaotic comb state under normal dispersion. It is suggested to briefly comment on or compare the properties of those chaotic microcomb states under different dispersion regimes for random bit generation.
3. The chaotic signal detection employing the silicon photonics WDM receiver was presented. While compared with the chaotic signal detected by commercial PD, the detected signal degenerated (Fig. 4). The supplementary information IX only shows the autocorrelation function of detected signals under different bias voltages. It is still not clear what degenerates the signal. Was the signal filtered or distorted due to the bandwidth or the saturation power? Why the channel numbers are different in Fig.4 f, g?

4. Considering the random number extraction from a chaotic or random process, a flatter probability distribution tends to be more favorable. Previous research has shown that the intensity probability distribution of the chaotic comb line could be fitted by the Rice distribution, where the distribution function shows a rapid peak. This is also shown in Fig. 3e. Is it possible to change the probability distribution?

5. Given the fact that the random bit stream relates to positive intensity values, the correlation data presented in Fig.3h should be scaled in the range from 0 to 1. This would help understand the cross-correlation between off-diagonal channels.

6. It is worth noting that some works (Iwami R, Mihana T, Kanno K, et al. Controlling chaotic itinerancy in laser dynamics for reinforcement learning[J]. Science Advances. 8(49), 2022) of optical decision using chaotic lasers show shorter convergence times even for problems with larger scales. Is the chaotic comb limit the convergence speed?

7. The claim “...platforms, helps to achieve the full width at half maximum (FWHM) of all comb lines smaller than 0.15 ns... indicating remarkable stability and inter-channel consistency’ is not clear. Necessary explanations or references need to be added.

8. What is the pump power conversion efficiency, in comparison with other integrated nonlinear platforms? Is the chaotic comb line power sufficient for the photodetection considering the future integration with on-chip PDs? Will the additional on-chip optical amplification corrupt the randomness or the distribution, instead of using off-chip EDFAs?

9. The authors used differentiation to artificially increase the sampling rate, make the distribution symmetric, and reduce the correlation, which is a widely used approach. The authors should explicitly introduce the idea of this approach with references.

Reviewer #3 (Remarks to the Author):

Review on “Massively parallel chaotic sources based on microcombs”

The manuscript written by B. Shen and co-authors presents the implementation of the random number generator based on the chaotic state of the microcomb. In this work, a highly nonlinear AlGaAsOI resonator with ~100 GHz free spectral range has been used for this purpose. The chaos-assisted random number generation has been applied to the multi-armed bandit (MAB) problem.

Despite the excellent presentation and quality of the presented data, the novelty of the manuscript is rather limited:

- 1) The chaotic state of the nonlinear resonator is rather a developed topic. Indeed, chaotic states of the microresonator have been achieved and demonstrated approximately 15 years ago. The very first demonstrated microcomb was found in this particular state. This state can be achieved even in the devices having large internal losses, mode crossings, and so, which would be an obstacle for the generation of coherent dissipative solitons. To my point of view all the results presented until section “Integrated parallel random bit generation” are almost completely covered by the three following papers [Coulibaly, et al. PRX 2019 {10.1103/PhysRevX.9.011054}, Wang, et al. Optics Express 2012 {10.1364/OE.20.029284}, Erkintalo, et al. Optics Letters 2014 {10.1364/OL.39.000283 }] as well as in Ref [11] of the manuscript.
- 2) Random number generator operating in the quantum regime has been implemented with integrated microresonators earlier with a different scheme [Okawachi, et al. Optics Letters 2016].
- 3) MAB problem parallelization has already been demonstrated in photonics for the 512 arm version [44].

In this regard, the title “Massively parallel chaotic sources based on microcombs” is too general and must be modified.

However, the combination of the tool, method, and problem is indeed novel and to my point of view can be considered for publication, if parts that represented the interest for the community are restructured and straightened.

The original part of the work is dedicated to the experimental realization of the random bit generator applied to the MAB demonstration seems to be interesting in the context of the emerging applications of the microresonator chaos. However, a detailed comparison with the existing methods must be improved and extended. As an example, the following questions can be answered:

1. Where is the place of the chaotic microcombs in the list of state-of-the-art methods according to the power budget, bandwidth, and autocorrelation? Several methods to generate chaotic optical signals must be compared.
2. In this regard, can the chaotic microcomb be considered as a competitive source for the random number generation?
3. Why does the MAB problems has been limited to 16 slots in the current work? What is the estimated capacity to scale this system?
4. What makes this chaotic microcomb different in comparison to all the prior devices demonstrated during last 15 years?

Minor remark:

Avoided mode crossings are very pronounced in the integrated dispersion profile as follows from Sup. Fig. 7. It would be beneficial for the reader to explore how the avoided mode crossings influence the properties of chaos by direct numerical simulations.

Concluding, I find that applications of the chaotic microcombs a promising direction of research with only a few existing works so far. However, the novelty of the presented data in this particular case is questionable. Thus, this manuscript can be re-considered for publication after major revision that must include removal of the data that does not have significant novelty, comparison of the device performance with existing photonic platforms, and clarification of the scalability of the system, ideally with experimental proof.

Remarks to the authors

The manuscript by Shen et al. reports an application of chaotic microcomb for random number generation.

The manuscript describes several key points:

1. Tuning into a chaotic state
2. Properties of the chaotic comb, including intensity noise correlations
3. RNG demonstration
4. Application to reinforcement learning

The reviewer has a few concerns, mainly:

- Refs [11,31,38] discuss the tuning into the chaotic comb and its properties in detail. However, the manuscript repeats the findings without referencing or comparison. The reviewer did not understand whether the authors brought new insights or repeated similar steps.
- Intensity noise correlations were studied in paper [1], with similar values reported by the authors.
- The authors claim: “As a result, so far in photonic integrated systems, the high performance chaotic function is still missing, which limits the scope of applications for integrated photonics. In this work, we overcome this long-standing problem by introducing another optical chaotic source, a chaotic microcomb 31–33, into photonic integrated circuits.”

Do the authors introduce a chaotic microcomb? Chaotic microcomb has been known for a decade, and **integrated chaotic microcomb has already been employed** in several applications Ref [11] and [2].

- Integrated chaotic source for RNG is the key result of this paper. Since the authors talk about its advantages and practicability, it would be good to see a comparison to the existing RNGs in terms of performance and power consumption.

The authors cite Ref [12], saying: “Despite their integration compatibility with CMOS electronics, these chaotic sources suffer from low bandwidth (on the order of hundreds of MHz [12])”. However, Ref [12] does mention chaotic electronic sources with 12.8 and 80Gs/s. Furthermore, Intel uses thermal noise to have a 3Gs/s random stream [3].

The authors demonstrate RNG at 6.67 Gs/s and 3 LSB. What is the limit there? Can you increase the sampling rate and/or number of LSB? How does it affect the ‘randomness’?

If the authors sample electronic noise of the scope (shut input), what would be the MAX rate that still produces random numbers?

What are the requirements for RNG in reinforcement learning? Do agents require random streams of 10Gs/s?

Please, find a list of extra comments below:

- One could use a semiconductor optical amplifier (can also be integrated) as an ASE source. This is another candidate for RNG, given ASE generators have been demonstrated [4]. Please, discuss potential benefits or drawbacks. ASE covers C band and doesn't require DEMUX units; it has higher conversion efficiency than combs.
- For the clarity of the manuscript, please, avoid non-scientific writing: "great orthogonality" (when does it stop being great?), "reliable chaotic properties", "indicates a good random bit extraction".
- Please, define the "effective bandwidth" term.
- Please, be more specific here:
 - "In addition, the broad chaos, starting from the very low-frequency baseband, holds consecutively strong components within its noise band, which is power-efficient for radio applications".
 - "The instantaneous intensity can reach an ultra-high level, which is a distinguishing feature of the spatiotemporal chaos... "
- Fig 3e – long tail can be seen in Log scale
- "A higher generation rate is obtained with the data down-sampled to 10 GSa/s, " – the sampling rate is increased.
- Fig 5e – why did the red trace fail to converge? Is this the average of over 10000 repetitions?
- Figures contain redundancies: parts of the experimental setup are repeated throughout all 5 figures. The intensity profile is repeated in Figs 1,2,3.
- Fig 3g – can be hardly seen with a current colormap, limits.
- Fig 3h – please, use Log-scale.

In its current form, I cannot recommend the manuscript for publication in Nature Communications.

[1] Marchand et al. "Soliton microcomb based spectral domain optical coherence tomography", NatureComm 2021.

[2] Ji, Xingchen, et al. "Chip-based frequency comb sources for optical coherence tomography." *Optics express* 27.14 (2019): 19896-19905.

[3] <https://www.intel.com/content/www/us/en/developer/articles/guide/intel-digital-random-number-generator-drng-software-implementation-guide.html>

3.2.1 Entropy Source (ES)

[4] Cao, Guangshuo, et al. "16.8 Tb/s true random number generator based on amplified spontaneous emission." *IEEE Photonics Technology Letters* 33.14 (2021): 699-702.

We appreciate the careful review by the reviewers and have modified the manuscript in accordance with their suggestions. Here, we present a point-by-point reply (in blue) to the reviewers' comments (in black), as well as the action taken (in red).

Response to the report from the Referee #1

Comments: *"The manuscript by Shen et al. reports an application of chaotic microcomb for random number generation.*

The manuscript describes several key points:

- 1. Tuning into a chaotic state*
- 2. Properties of the chaotic comb, including intensity noise correlations*
- 3. RNG demonstration*
- 4. Application to reinforcement learning"*

Our reply: We appreciate the reviewer for carefully reviewing our manuscript and for their helpful comments. These comments, followed by our point-by-point responses, are shown below.

Comment 1: "The reviewer has a few concerns, mainly:

Refs [11,31,38] discuss the tuning into the chaotic comb and its properties in detail. However, the manuscript repeats the findings without referencing or comparison. The reviewer did not understand whether the authors brought new insights or repeated similar steps."

Our reply: It is true that the tuning into the chaotic comb has been discussed in previous works. In the revised manuscript, the first part of result, "Route into the chaotic microcomb", is moved to supplementary note III.

Comment 2: "Intensity noise correlations were studied in paper [1], with similar values reported by the authors."

Our reply: Yes, the intensity noise correlation was presented in paper [Nature Communications, 12, 427 (2021)]. However, it is worth noting that this paper [1] **only valued the correlation between two certain channels**. For a spatiotemporal or a multi-output chaotic system, the general inter-channel correlation cannot be easily captured by calculating the correlation between certain channels. For instance, a periodic inter-channel correlation was presented in [Physical review letters, 77, 4162-4165 (1996)]. **It is still unclear whether the chaotic signals carried by different comb lines of chaotic combs are correlated with other lines.**

In this work, as far as we know, **the inter-channel correlation between every pair of comb lines is investigated for the first time**, both in experiment and in simulation. Two key properties of the inter-channel correlation are revealed: **1) symmetrically distributed comb line pairs exhibit positive correlation, which was not previously reported due to the limited investigation of certain channel pairs** [Nature Communications, 12, 427 (2021)]; and **2) for other comb line pairs around the pump mode, the inter-channel correlation can be lower than 0.04**. Our discovery indicates that it would be a proper choice to use comb lines on the same side of the pump mode as parallel chaotic sources. Low inter-channel correlation is significant for applications ranging from private communication and key distribution to chaotic lidar and radar. In the private communication and key distribution, low inter-

channel correlation is crucial for ensuring the security and confidentiality of the information being transmitted. In these systems, the chaotic source is used to generate a random key that is shared between the communicating parties. The independence of each channel are essential to ensure that the key cannot be easily intercepted or decoded. As for chaotic radar/liar system, low inter-channel correlation helps to reduce interference and improve the accuracy of the system. In these systems, chaotic sources are used to generate a sequence of chaotic signals that are transmitted and received by the radar/lidar. The low correlation between channels helps to reduce interference from other channels and improve the accuracy of the system by increasing the signal-to-noise ratio. Moreover, the uncorrelated channels could further help reduce the number of the receivers by means of co-detection, thus decreasing the complexity and power consumption of an integrated parallel systems [Nature Photonics, DOI:10.1038/s41566-023-01158-4 (2023)].

Comment 3: “The authors claim: “As a result, so far in photonic integrated systems, the high performance chaotic function is still missing, which limits the scope of applications for integrated photonics. In this work, we overcome this long-standing problem by introducing another optical chaotic source, a chaotic microcomb 31–33, into photonic integrated circuits.” Do the authors introduce a chaotic microcomb? Chaotic microcomb has been known for a decade, and integrated chaotic microcomb has already been employed in several applications Ref [11] and [2].”

Our reply: Sorry for the misleading. We agree with the referee that the chaotic comb has been known for a decade and has already been employed in several applications. In this work, **we introduce the chaotic comb as a massively parallel chaotic source, rather than “introducing” or “discovering” it.** Please find below the clarification on these points.

1. **The thorough characterization of chaotic combs:** Two questions must be answered when using chaotic combs as parallel chaos sources. The first question is **whether the chaotic performance of each tooth of the chaotic combs is suitable for the intended application.** The second question is **whether there is any correlation between different comb teeth.** For most previous works such as [Physical Review X, 9, 011054 (2019)], the whole chaotic comb, not the characteristic of each single line of it, was studied. In ref [11], the bandwidth and full-width-half-max of autocorrelation of each comb line were valued, while the inter-channel correlation was not studied. In [Nature Communications, 12, 427 (2021)], the correlation between only two certain channels was examined. In this paper, **we conduct a thorough characterization of the chaotic optical comb.** Through experiments, we characterized the autocorrelation in time domain of all comb teeth in C band. The comb teeth near the pump mode show close chaotic performance. And the power of each comb tooth is similar. Furthermore, we provide, for the first time, the inter-channel correlation between each pair of comb lines, as discussed in “**reply to comment 2**”.

2. **The outstanding performance of chaotic combs based on AlGaAsOI:** It is worth noting that we can obtain superior chaotic optical combs with moderate pump power, due to the excellent nonlinear effect of AlGaAs. Compared to the radio frequency spectrum width of all integrated chaotic optical combs previously reported, the chaos bandwidth generated by using AlGaAs ring is the widest. It is important to note that the performance of our current optical combs is limited by the carrier absorption effect. We expect to achieve a chaotic optical comb with higher performance by reducing carrier lifetime and minimizing the effect of carrier absorption through structures such as PIN. Moreover, AlGaAsOI microrings heterogeneously integrated with the SOI platform have been reported in [Photonics

Research, 10, 02000535 (2022)], and the corresponding quality factor is up to 1.12×10^6 , which is sufficient to support the excitation of chaotic optical combs. By combining high power on-chip laser, high nonlinear microring and silicon photonics, we hope to realize a **fully integrated parallel chaotic source generation and processing chip**, as we present in Fig. 1.

Based on the above two advantages, the chaotic optical comb can be used as an integrated large-scale parallel chaotic source. In terms of application, although chaotic optical combs are used as broadband source in [2], where the characteristics of chaotic signals are not utilized. In [11], the chaotic optical comb is used as a parallel source. Here, the chaotic optical comb is used as a parallel source of chaos. **Such parallelism requires strict inter-channel uncorrelation.** This low correlation of signals between channels has important applications in many fields, as we discussed in "*reply to comment 2*".

In terms of single-channel performance, we must acknowledge that the current single-channel performance of chaotic optical combs is inferior to that of chaotic lasers in terms of single-channel power and chaotic bandwidth. Therefore, further explorations for higher bandwidth parallel chaos generation are discussed in our revised manuscript. By introducing PIN structures to reduce carrier lifetime and carrier absorption, single-channel performance comparable with chaotic laser performance can be obtained. Also, the power of each comb tooth can be improved by designing the coupling coefficient to improve the conversion efficiency of the optical comb. In this paper, to compensate for the disadvantage of chaotic bandwidth, we also attempt to use the scheme of combining two optical combs, inspired by the scheme of chaotic lasers [Optics Express, 28, 3686 (2020)]. The method has led to a considerable radio frequency bandwidth increasing and achieve 120 Gbps random bit generation in a single channel (30 Gbps in the previous version), as shown in "*reply to comment 6*".

In addition, we would like to discuss the implementation of parallel optical chaotic sources. We will not discuss electrical chaos schemes here, due to their relatively low chaos bandwidth, as discussed in "*reply to comment 5*". In Fig. 1, we show the generation schemes of parallel chaotic sources including the chaotic optical comb. One scheme is based on chaotic lasers, where a specific feedback loop with a laser diode is used for per channel. In order to obtain parallel output, multiple semiconductor lasers are needed. Compared with optical chaotic combs, the cost for packaging and control of laser array is much higher, similar to the comparison between the soliton microcomb and the laser array for optical communication, especially when considering the need for signal transmission using wavelength-division multiplexing technology.

In conclusion, in this paper, we propose and verify that **chaotic optical combs can serve as integrated large-scale parallel chaotic optical sources**. Compared with previous optical chaotic sources, chaotic optical combs have **the advantages of low cost, large scale and integration**. Compared with previous studies on chaotic optical combs, this paper **presents a thorough performance characterization of chaotic optical combs**, especially the characterization of the correlation between channels, which is the necessary guarantee of "parallelism". This characterization expands upon previous studies of chaotic optical combs.

To make it clearer, we have added a review of previous reported studies about chaotic combs in the Introduction part:

(Main part, page 2, line 17) **The use of a chaotic comb as a parallel chaotic source offers a promising alternative, as spatiotemporal chaos can be achieved by pumping an optical nonlinear microcavity with sufficient power. In frequency domain, the comb is consisted of well-separated comb lines, each**

exhibiting poor coherence [31–33]. This promises tremendous parallelism provided by optics through wavelength division-multiplexing technology. Recently, chaotic combs have been employed as parallel optical sources for optical coherence tomography [34] and parallel ranging [11], but the parallelization was based on frequency domain rather than on the chaotic signal carried by each channel. For parallel chaotic sources, channels should be uncorrelated with each other, which is vital to applications ranging from private communication [4,35] and key distribution [5] to chaotic lidar/radar systems [7,36]. However, it is still unclear whether chaotic combs can function as parallel chaotic sources. In this work, we fill this gap by providing a full investigation of chaotic comb lines, and introduce the optical chaotic comb as a massively parallel chaotic source for photonic integrated circuits.

Comment 4: “Integrated chaotic source for RNG is the key result of this paper. Since the authors talk about its advantages and practicability, it would be good to see a comparison to the existing RNGs in terms of performance and power consumption.”

Our reply: Thanks for your suggestion. In the revised manuscript, a table for the comparison of different RNGs and optical chaotic sources are added. While the comparison of power consumption is not included in the table due to the lack of data in reported paper. A discussion of the power consumption is given.

(Main part, page 5, line 29) In Table I, we present a comparison of different random bit generation schemes. Due to the massively parallelism provided by chaotic combs, the total bit rate of chaotic-comb-based RBG shows the highest generation rate among optical-chaos-based methods. As the lack of data in published articles, the power consumption is not listed in Table I. Considering simple setup with one pump laser and one microcavity, the generation of optical chaotic sources based on chaotic combs is power efficient and low-cost compared with chaotic lasers [52,53], which require at least one laser diode per channel. In addition, our work gives the first demonstration of parallel chaotic signal detection employing the integrated SiPh chip. Although the generation rate of chaotic-comb based RBG is lower than ASE (amplified spontaneous emission)-based RBG with spectral or spatial parallelism, the use of chaotic systems offers the ability to synchronize the output of two chaos generators [4,35], which can be arranged at the transmitter and receiver sides respectively. This synchronization enables key distribution and private communication, which is not possible with stochastic sources such as ASE and random lasers.

Table 1. Comparison of different schemes of random bit generation based on optics

Scheme	Total bit rate (Gbps)	One channel (Gbps)	Channel number	Bandwidth (GHz)	Integrated source	Integrated processor
Random laser ⁴²	540	540	1	29 +	N	N
Quantum phase fluctuation of laser ⁴³	117	117	1	~	N	N
Bi-phase State of OPO ⁴⁴	0.002	0.002	1	~	Y	N
Vacuum state fluctuation ⁴⁵	18.8	18.8	1	~2.3+	Y	Y
ASE ⁴⁶	16800	400	42	<40*	N	N

Laser diode (ASE) ⁴⁰	254000	2000	127	315*	Y	N
Chaotic laser ⁴⁷	320	320	1	16.7*	N	N
Chaotic laser ⁴⁸	1200	600	2	26*	N	N
Chaotic laser ⁴⁹	2240	320	7 (3)**	~	N	N
Chaotic laser ⁵⁰	10	10	1	11.6 †	Y	N
Chaotic laser ⁵⁴	21.1	21.1	1	7†	Y	N
Chaotic comb	300	20	15	5.6†	Y	Y
(this work)	3840	120	32	9.6†	Y	N

*Effective bandwidth, defined as the width of spectral segment that accounts for 80% of the total power.

†10-dB bandwidth.

**7 channels of random number generation, by linear combining 3 channels of optical chaotic signals.

Comment 5: “The authors cite Ref [12], saying: “Despite their integration compatibility with CMOS electronics, these chaotic sources suffer from low bandwidth (on the order of hundreds of MHz [12])”. However, Ref [12] does mention chaotic electronic sources with 12.8 and 80Gs/s. Furthermore, Intel uses thermal noise to have a 3Gs/s random stream [3].”

Our reply: The hundreds of MHz bandwidth and 80 Gbps random bit generation rate are not contradictory. Fig. R1 shows the radio frequency spectrum of the chaotic source proposed in [Physical Review Letters, 111, 044102 (2013)]. The bandwidth was on the order of hundreds of MHz. Employing the pure electronics, several chaotic or stochastic sources have to be combined to realize a high speed RNG. In [Physical Review Letters, 111, 044102 (2013)], 6 independent sources were combined to reach the RNG rate of 80 Gbps.

Fig. R1 The radio frequency spectrum of the chaotic source proposed in [12].

Also, it has to be clarified that the RNG rate could be several times larger than the radio frequency bandwidth. For example, in this work, the 10 dB bandwidth of the comb line is 5.6 GHz, while the RNG rate could reach 20 Gbps employing the integrated Ge-Si photodiode.

For an unambiguous expression, the review of electronics is modified in the revised manuscript:

(Main part, page 1, line 67) Despite their integration compatibility with CMOS electronics, these chaotic sources suffer from low bandwidth (on the order of hundreds of MHz [12]) and the combination of several sources is necessary for high throughput rate, which limit the electronic processing speed and therefore lagging the system performance.

Comment 6: “The authors demonstrate RNG at 6.67 Gs/s and 3 LSB. What is the limit there? Can you increase the sampling rate and/or number of LSB? How does it affect the ‘randomness’?”

Our reply: For an entropy source, the max generation rate of random bit should be limited by the entropy rate. Here, we calculated the Cohen-Procaccia entropy $h_{cp}(\varepsilon, \tau, d)$ employing the Cohen-Procaccia algorithm [APL Photonics, 2, 090901 (2017)]. ε is given as $\frac{I_{max}-I_{min}}{2^N}$, where N is the number of digital, I_{max} (I_{min}) is the maximum (minimum) of the recorded raw data I . τ is the sampling period. d is the embedding dimension and is set to 3.

Fig. R2 Cohen-Procaccia entropy rate of recorded data

As shown in Fig. R2a, first, the entropy rate per sample increases rapidly with the increase of the sampling period, and then tends to level off as sampling period reach 150 ps. Thus, in the main text, we down-sample the raw data to 6.67 Gs/s for random bit generation. At the sampling period of 150 ps, we calculate the entropy rate under different numbers of digital. It is worth noting that the Cohen-Procaccia entropy rate is about 23 Gbps, closing to the generation rate demonstrated in the main text. In Fig. R3, we give the comparison of random bit sequences generated with different numbers of LSB. As the number of LSBs increases to 4, the distribution is not flat anymore, indicating that some bit sequences (for example, 0111) would appear more frequently than other patterns.

Fig. R3 Frequencies of different bit patterns for different numbers of LSBs

It is possible to get a higher generation rate with more complex post-processing, such as reshaping the raw data with certain map [IEEE Photonics Technology Letters, 33, 699 (2021)], but it is questionable whether it is true random as post-processing algorithms cannot increase the entropy production rate of the physical source.

Except for the random number generation using every comb line presented in the original manuscript, we give another method to push the throughput rate to a higher level. Inspired by chaotic lasers where two chaotic lasers could be mixed to generate a complex signal with larger bandwidth compared with raw chaotic lasers [Optics Express, 25, 3153-3164 (2017)], we adopt two chaotic combs here. The

frequency of each comb is slightly different with comparatively close FSR. By selecting and then injecting two comb lines from different chaotic combs into a photodiode, a broad RF spectrum could be obtained. The corresponding result are added to the revised manuscript:

(Main part, page 5, line 7) The generation rate of single chaotic comb is mainly limited by the chaotic bandwidth, which is degenerated by the nonlinear absorption. To compensate this, here we provide a method by a dual-comb scheme. The schematic is shown is Fig. 3c, where two combs are pumped by two lasers with a frequency difference around 4 GHz. In experiment, two microcavities are packaged with temperature controllers and the chaotic combs are generated by sweeping the resonant frequency by tuning the chip temperature, while the pump lasers are kept fixed. The comb teeth of the two chaotic combs are filtered, amplified and combined, before being sent to the photodiode. The time domain waveforms of each tooth pairs are recorded at a sampling rate of 80 GSa/s. The raw data are down-sampling to 40 GSa/s and processed as described above. In this case, the single channel generation rate can be increased to 120 Gbps, which is comparable to the novel random bit generator based on chaotic lasers. 32 channels on the same side of the pump mode are recorded, processed, and successfully pass the NIST SP 800-22 test (see in Fig. 3h), corresponding to an aggregation rate of 3.84 Tbps.

Fig. 3. The parallel random bit generator based on a microresonator and a SOI chip. a, Optical microscope photograph of the SiPh receiver. b, The set-up scheme for parallel random number generation; AWG, arrayed waveguide grating; PD, photodetector; OSC, oscillator; D, delay unit; XOR, exclusive-OR. c, The setup for random bit generation using two chaotic combs. d, The possibility distribution function of the differential data. e, The distribution of the extracted 3 LSBs. f, The ACF of the generated bit sequence. The red line indicates the lower limit determined by $1/\sqrt{n}$. The NIST SP800-22 test results for signals detected by SOI PD with setup shown in (b) (g) and commercial InP PD with

setup shown in c (h).

Comment 7: “If the authors sample electronic noise of the scope (shut input), what would be the MAX rate that still produces random numbers?”

Our reply: It is a quite interesting question. Actually, one possible method to obtain random bit sequence is to take the last bit of the original sampled data. To answer the question, we recorded the data produced by the scope without any external input signal. Due to the weak amplitude of the sample electronic noise (± 2 mV in experiment), the sampled maximum voltage is set to 2 mV, which is smallest for the scope employed here. The noise is recorded at 80 GSa/s and down-sampled to different sampling rate. The least-significant bit extraction is employed to the down-sampled data. Fig. R4 shows the autocorrelation function and radio-frequency noise of the recorded data.

Fig. R4 The autocorrelation function (a) and the radio frequency spectrum (b) of recorded data from scope without input.

As for the autocorrelation function, a periodic oscillation could be found. The periodic oscillation indicates that a periodic trench exists in the recorded data. The periodic trench could be figured out obviously in the radio frequency domain, where three peaks located at 10 GHz, 20 GHz and 30 GHz could be found. The obvious peaks are correlated with the internal clock signal. Such period trench is unwanted for random bit generation.

Fig. R5 The distribution of raw recorded data (a), extracted 4 LSB (b1), 3 LSB(b2), 2 LSB (b3) and 1 LSB (b4).

Fig. R5a shows the distribution of the raw recorded data. Due to the weak amplitude, the recorded data

concentrated on tens of different values. This blocks the extraction of random bits from raw data. As shown in Fig. R5b, even for the case of that one bit extracted from one sampled data, at least one number shows a higher frequency rate compared with other numbers. This indicate that certain sequence appears more frequently, which is not a “random” sequence.

Thus, in practice, we have not extracted random bit sequences that could pass the NIST tests successfully. The test results are presented below:

Table. R1 NIST SP800-22 test results for random bit sequences generated by the electronic noise of the scope

Number Rate (GSa/s)	11	13	16	20	27	40	80	80	80
Number of Extracted bits	1	1	1	1	1	1	1	2	3
Frequency	P	P	F	F	P	F	F	F	F
Block frequency	F	P	P	F	P	P	F	F	P
Cumulative sums	F	F	F	F	F	F	F	F	F
Runs	F	F	F	P	P	F	F	P	P
Longest run	P	P	P	P	P	P	F	P	P
Rank	P	P	P	P	P	P	P	P	P
FFT	P	P	P	P	P	P	P	P	P
Nonoverlapping template	F	F	F	F	F	F	F	F	F
Overlapping template	P	P	P	P	P	P	P	P	P
Universal	P	P	P	P	P	P	P	P	P
Approximate entropy	P	P	F	F	F	F	F	F	P
Random excursions	P	P	P	P	P	P	P	P	P
Random excursions variant	P	P	P	P	P	P	P	P	P
Serial	P	F	F	F	P	F	F	F	P
Linear complexity	P	P	P	P	P	P	P	P	P

“P” and shadowed with green marks successfully passing the test

“F” and shadowed with red marks unsuccessfully passing the test

In addition, we have to point out that the influence of electronic noise is unwanted for random bit generation based on optics. Compared with electric, “optical systems are especially well-suited for random number generation due to resistance to external interference, speed, and access to quantum mechanical processes”, which is commented in [APL Photonics, 2, 090901 (2017)]. For the study of random bit generation based on physical sources, the influence of electronic noise from digitizer has to be avoided. Our aim is to study the random number performance generated by the entropy source itself. For example, considering part of the entropy extracted from random numbers comes from electrical noise, this will hinder some potential applications of random numbers. If the optical chaotic source based on synchronization is considered for key distribution, entropy based on electrical noise cannot achieve synchronization, destroying the performance of key synchronization at both ends.

Comment 8: “What are the requirements for RNG in reinforcement learning? Do agents require random streams of 10Gs/s?”

Our reply: These are quite interesting questions. Actually, for present reinforcement learning developed in digital computers, pseudo-random number generators are sufficient for the purposes of exploration, environment simulation and initialization. Using pseudo-random number generators, based on the

same random seed and algorithm, one could reproduce the random sequence, helping the examination and comparison. As shown in [https://arxiv.org/pdf/1910.06591.pdf], the training speed of the platform could reach 2.4 MFPS (frames per second), indicating a low demand of generation rate.

While in this work and other papers focusing on the chaos-based decision-making accelerators, **the aim is to pursue better performance by employing specialized computing architectures based on optical chaos, compared with conventional computer-based algorithm.** In revised manuscript, we verify the ability of chaotic combs as parallel chaotic sources to enhance the decision-making accelerator. The chaotic-comb-based scheme shows faster convergence speed for large-scale problems than widely-used computer-based algorithms. Despite weaker scalability of frequency domain (1-dimension space) compared with spatial domain (2-dimension space), the chaotic-comb-based scheme benefits from the integrated optoelectronics, providing a compact, low-cost, mass-producible decision marker meanwhile hold the possibility for further co-integration with electronic processors. Also, the time-division-multiplexing could be applied to enlarge the scale, contributing to a faster variation of chaotic combs compared with spatial chaos systems.

(Main part, page 6, line 22) **Optical chaos, with its fast and complex internal time evolution, is a powerful entropy source that can be used for exploration purposes [54].** Chaotic lasers have been successfully employed to solve the multi-armed bandit problem (MAB) [29,55,56], a fundamental problem of reinforcement learning. **To scale up the problem exponentially, a parallel scheme is required that employs parallel chaotic signals or is based on time-division-multiplexing [57].** In this section, we propose the use of the chaotic comb as a massively parallel chaotic source to solve the MAB. Fig. 4a shows the schematic of the optical decision making. There are N slot machines with a reward possibility $p_i (i = 0 \sim N)$ respectively. The decision maker needs to find out the slot machine with the highest reward possibility by consecutively playing the slot machines. For each play, the decision maker chooses and plays one slot machine. The decision maker will change its selection strategy based on whether a reward is obtained. In the decision maker, N chaotic signals $\Delta I_i(t) (i = 0, 1, 2, 3)$ are detected and add with bias values $B_i(t)$ respectively to produce $A_i(t)$. Each channel is correlated with one slot machine. For each play, the slot machine with the highest biased signal $A_i(t)$ is selected and played. As continuously playing the game, bias values will be tuned based on tug-of-war method [56] (see Methods).

In experiment, we detect 32 channels at the short wavelength side of the pump mode one by one, to solve the 32-armed bandit problem. The hit probabilities of the 32 slots are set as follows: $P_1 = 0.7, P_2 = 0.5, P_3 = 0.9, P_4 = 0.1, \dots, P_{2j-1} = 0.7$ and $P_{2j} = 0.5$. As shown in Fig. 4b, the chaos-based decision maker would initially explore a wide range to identify the slot machine with the highest hit probability. After sufficient exploration, the slot machine 3 is identified as the best one and frequently selected, indicating successful decision making. To evaluate the performance of the decision making based on chaotic combs, the slot machines are played $C = 1000$ cycles consecutively, and this process is repeated $T = 1000$ times. For t -cycle, the correct decision ratio (CDR) is defined as T_{hit}/T , where T_{hit} represents the number of times a reward is obtained at the t -th cycle among the T processes. A CDR above 95% is viewed as a successful decision and the first cycle getting a CDR > 95% is defined as the convergence cycle (CC). Fig. 4c shows the CDR evolution along with the increase of cycles. Under the same system condition, the CDR evolution remains consistent across different N_{target} , which represent the number of the slot with the highest hit probability.

The scalability is a critical characteristic of the decision maker, where the convergence cycle will exponentially increase as the rise of problem scales. To assess the ability of solving large scale problems,

we use the chaotic comb to solve the MBA problems with the number of slot machines $N = 4, 8, 16, 32, 64, 128, 256$. Limited by the gain bandwidth and the spectrum width of the chaotic comb, the number of available channels is 32. For $N > 32$, we record the 32 channels repeatedly to obtain enough channels of chaotic signals. As shown in Fig. 4d, the chaotic-comb-based decision marker can solve the MBA problems even with a large scale. Fig. 4e gives the performance comparison between chaotic-comb-based decision marker and other methods [56]. Fitted with the power function, the relationship between the convergence cycle CC and the number of slot machines N can be described as $CC = 26.48 \cdot N^{0.89}$. Compared with widely-used algorithm such as UCB1-tuned (upper confidence bound 1-tuned) algorithm and Thompson sampling algorithm, methods based on optical-chaos show smaller scaling exponents, suitable for large-scale problems. Among the optical chaos, the decision maker shows a superb convergence speed, close to the spatial-chaos based method. While compared with the spatial chaos system, the chaotic microcomb benefits from integrated optoelectronics, providing a compact, low-cost, mass-producible decision marker. Despite having fewer channels compared to the spatial chaos system [56], the chaotic comb feathers high generation rate (40 GSa/s employed here for decision making) and the adoption of time-multiplexing [57] can be a promising approach to scale up the system, even though it may result in a tradeoff between speed and scalability.

Fig. 4 Multi-armed bandit problem solving based on chaotic combs. a, The scheme of the optical decision making based on the parallel chaotic source. b, One decision process for 32-armed bandit problem. The left figure shows the hit probability distribution of 32 slots, where the 3rd slot has the highest hit probability 0.9. c, the evolution of corrected decision rate with the increase of cycles. The red dashed line marks the corrected decision rate of 95%. d, The evolution of corrected decision rate under different scales. e, the comparison of scalability between chaotic-comb-based decision maker and other methods.

In addition, we find the expression of the decision-making accelerator in the Abstract might be misleading, so the description is re-modified as follows:

(Abstract, page 1, line 38) Driven by such a source, we demonstrate the first parallel chaotic systems in integrated photonics: 15-channel random bit generation with 20 Gbps bit channel rate and scalable decision-making accelerator for multi-armed bandit problems.

Comment 9: "Please, find a list of extra comments below:

One could use a semiconductor optical amplifier (can also be integrated) as an ASE source. This is another candidate for RNG, given ASE generators have been demonstrated [4]. Please, discuss potential benefits or drawbacks. ASE covers C band and doesn't require DEMUX units; it has higher conversion efficiency than combs."

Our reply: It is true that the ASE generator or SLD (super-luminescent diode) would be a competitive entropy source for random bit generation, considering the simple setup, high radio frequency bandwidth and the ability of parallelism. As shown in the table for comparison of different random bit generators, the highest output rate is based on ASE combined with space division multiplexing. While we have to point out that the high output rate is not the only demand for applications of random bits. Compared with ASE source, one of the most significant properties of optical chaotic source is the ability of chaotic synchronization. By constructing a pair of synchronized chaotic sources, both sides can share key information through synchronized chaos, so as to realize key distribution. In addition, secure communication based on chaotic synchronization, such as image encryption, cannot be realized directly through ASE source. Therefore, chaos source has advantages over ASE source in potential application prospect.

Furthermore, compared to the ASE scheme, the optical comb scheme offers superior scalability in the wavelength domain. By designing the microcavity dispersion to be weakly anomalous, the number of channels can be substantially increased. Due to the optical comb's gain originating from the optical nonlinear effect, it is not restricted by the gain bandwidth of the material, enabling it to achieve a wider spectrum and greater scalability.

In addition to being an entropy source, random sequences are widely used in radar/Lidar and other fields. In the radar application scenario, limited by the bandwidth of optoelectronic components, the actual bandwidth used is often only at the level of GHz. For an ultra-broadband ASE source, a low-pass filter is arranged due to the limited bandwidth of optoelectronic components, resulting energy loss at high frequency thus lowering the efficiency. For example, the bandwidth of the commercial photodiode APD (https://www.thorlabs.com/newgrouppage9.cfm?objectgroup_id=3010) is 400 MHz.

Comment 10: "For the clarity of the manuscript, please, avoid non-scientific writing: "great orthogonality" (when does it stop being great?), "reliable chaotic properties", "indicates a good random bit extraction"."

Our reply: The reviewer's suggestion is highly appreciated. We have revised the manuscript accordingly to use more specific and quantifiable language. Instead of "great orthogonality", we will use specific values to describe the orthogonality. The requirement of inter-channel correlation varies for different applications. For example, in [Science, 371, 948 (2021)], the correlation between raw data of adjacent channels is about 0.15 while it is enough for parallel random bit generation thanks to the post-processing algorithms. In [Physical review letters, 77, 4162 (1996)], the chaotic map with inter-channel correlation around 0.1 is studied for multichannel secure communication. For chaotic radar/lidar, the inter-channel correlation should be as low as possible. In [Optics Express, 30, 4782 (2022)], the accuracy inter-channel correlation is not given while the inter-channel correlation is estimated to be around 0.03 from figures. In general, the 0.04 inter-channel correlation achieved here is low enough for present applications. The modified expressions are listed below:

(Main part, page 4, line 65) Except for the symmetrical modes, the correlation between comb lines is lower than 0.04, which is low enough for applications require parallel chaotic sources [7,35,36,40].

The "reliable chaotic properties" has been removed in the revised manuscript.

(Main part, page 4, line 69) Hundreds of channels, the uniform power distribution, the negligible cross-correlation between different channels, and the comparable chaotic bandwidth with chaotic lasers, ~~and the reliable chaotic properties ensured by the nature of spatiotemporal chaos~~, all these properties suggest the chaotic comb is an attractive massively parallel chaotic source.

The "good random bit extraction" has been replaced with proper description.

(Main part, page 4, line 110) A normal distribution of the extracted 3 LSBs (Fig. 3e) indicates that the generated bit sequence contains different bit patterns with equal frequencies.

Comment 11: "Please, define the "effective bandwidth" term."

Our reply: Sorry for improper writing. For the value of bandwidth of chaotic signal, there is various definition for bandwidth. In [IEEE Journal of Quantum Electronics, 48, 1010 (2012)], the spectral segment that accounts for 80% of the total power is defined as effective bandwidth. In [Scientific Reports, 11, 13182 (2021)], the 10-dB bandwidth is used to value the bandwidth. In this paper, we adopt the 10-dB bandwidth to value the chaotic bandwidth. Corresponding description is given in the Supplementary note I.

Comment 12: "Please, be more specific here:

o "In addition, the broad chaos, starting from the very low-frequency baseband, holds consecutively strong components within its noise band, which is power-efficient for radio applications".

o " The instantaneous intensity can reach an ultra-high level, which is a distinguishing feature of the spatiotemporal chaos... ""

Our reply: We apologize for the vague description in the text. The first sentence was intended to compare the power distribution at low frequencies between chaotic lasers and chaotic optical combs, where the power distribution at low frequency is relatively low for chaotic lasers. For the conversion from optical signal to electrical signal, a photodetector is used which express a low-pass response. Thus the raw output of a chaotic laser is inefficient as discussed in [Optics express, 26, 12230 (2018)]. For chaotic combs, it shows consecutively strong components within its noise band, which is power-efficient for conversion to the electrical signal.

(Main part, page 4, line 7) In addition, the broad chaos, starting from the very low-frequency baseband, holds consecutively strong components within its noise band, which is power-efficient for radio applications considering a low-pass response in the optical receiver for RF signal generation [7].

For the second sentence, it was for comparation with other microcomb state, where the time domain variation shows a constant or period trajectory. Considering this sentence expressed similar meaning with the last one, it is deleted in the revised manuscript.

Comment 13: “Fig 3e – long tail can be seen in Log scale.”

Our reply: Thanks for your suggestion. The y-axis has been changed to log scale in the revised manuscript, which you can find in “reply to comment 18”.

Comment 14: “A higher generation rate is obtained with the data down-sampled to 10 GSa/s, – the sampling rate is increased.”

Our reply: Sorry for unclear description. Two different sampling rates are presented in the original manuscript, 6.67 GSa/s for integrated Ge-Si photodiode and 10 GSa/s for commercial InP photodiode. The “10 GSa/s” are actually used for random bit extraction using data detected by the commercial InP photodiode.

In the revised manuscript, the description of the random bit generation using the commercial photodetector and one chaotic comb is moved to the Supplementary note X.

Comment 15: “Fig 5e – why did the red trace fail to converge? Is this the average of over 10000 repetitions?”

Our reply: Yes, the red trace in Fig. 5e is averaged over 10000 repetitions. In the original manuscript, the MAB experiment using pseudorandom sequences produced by RAND function of Matlab is carried out by time division method. One pseudorandom sequence $I_{raw}(i)$ is produced and divided into four channels I_0 , I_1 , I_2 and I_3 , where $I_m(n) = I_{raw}(4n + m)$ ($m = 0,1,2,3$ and $n = 1,2,3, \dots$). The reason for the failure convergence is the limited diffusivity of pseudorandom sequences produced by RAND function as revealed in [Scientific Reports, 8, 10890 (2018)]. In the revised manuscript, we redo the experiments for the MBA problem. And the results in the original manuscript are removed.

Comment 16: “Figures contain redundancies: parts of the experimental setup are repeated throughout all 5 figures. The intensity profile is repeated in Figs 1,2,3.”

Our reply: Thanks for your suggestion. In the revised manuscript, the setup for comb generation and amplification is marked as “Parallel chaotic source”, as shown in revised Fig. 2b in “reply to comment 18”. In revised Fig. 3b and c, the same setup is depicted as “Parallel chaotic source” as shown in “reply to comment 6”. In the revised manuscript, the intensity profile is shown in Fig. 1 and Fig. 2. The intensity profile shown in Fig. 1 is a schematic representation illustrating the difference between the input and output fields, which is not the experimental recorded data. Thus, the intensity profile is retained in the revised Fig. 2, which presents the experiment result.

Comment 17: “Fig 3g – can be hardly seen with a current colormap, limits.”

Our reply: Thanks for your suggestion. The revised figure is presented in “reply to comment 18”.

Comment 18: “Fig 3h – please, use Log-scale.”

Our reply: Thanks for your suggestion. The revised figure is shown below:

Comment 19: “In its current form, I cannot recommend the manuscript for publication in Nature Communications.”

Our reply: Thanks again for carefully reviewing our manuscript and for helpful comments. We hope that our revisions have addressed your concerns, and we look forward to your review of our revised manuscript.

Response to the report from the Referee #2

Comments: “In this manuscript, the authors report a demonstration of a chaotic generator that can provide up to 15 channel random bits at high speed, based on a compact AlGaAsOI integrated photonic nonlinear microresonator along with a silicon photodetector array. These chip-based chaotic sources were applied to potentially important applications such as parallel random bit generation and computation acceleration, supported with concrete and comprehensive experimental demonstrations. Overall, I find the results novel and believable, and the manuscript is well-written. Although the chaotic optical combs in microresonators have been reported (the authors have cited prior work), the large-scale parallelization of the random bit generation and its application in bandit problem-solving using

integrated photonic devices will be of interest to a broad audience. For this reason, the presented results warrant the publication of this work provided addressing the following comments:

Our reply: We thank the reviewer’s positive comments on the novelty of our work. We made the point-by-point response according to your comments as follows.

Comment 1: “To clearly position the breakthrough claimed in this work, a table that summarizes the state-of-the-art chaotic sources, in particular for those using photonics, is suggested to be provided in the main manuscript.”

Our reply: Thanks for the suggestion. In the revised manuscript, a table that summarizes random bit generators and optical chaotic sources is added at the end of the “Integrated parallel random bit generation” part, with detailed discussions.

(Main part, page 5, line 29) In Table I, we present a comparison of different random bit generation schemes. Due to the massively parallelism provided by chaotic combs, the total bit rate of chaotic-comb-based RBG shows the highest generation rate among optical-chaos-based methods. As the lack of data in published articles, the power consumption is not listed in Table I. Considering simple setup with one pump laser and one microcavity, the generation of optical chaotic sources based on chaotic combs is power efficient and low-cost compared with chaotic lasers [52,53], which require at least one laser diode per channel. In addition, our work gives the first demonstration of parallel chaotic signal detection employing the integrated SiPh chip. Although the generation rate of chaotic-comb based RBG is lower than ASE (amplified spontaneous emission)-based RBG with spectral or spatial parallelism, the use of chaotic systems offers the ability to synchronize the output of two chaos generators [4,35], which can be arranged at the transmitter and receiver sides respectively. This synchronization enables key distribution and private communication, which is not possible with stochastic sources such as ASE and random lasers.

Table 1. Comparison of different schemes of random bit generation based on optics

Scheme	Total bit rate (Gbps)	One channel (Gbps)	Channel number	Bandwidth (GHz)	Integrated source	Integrated processor
Random laser ⁴²	540	540	1	29 †	N	N
Quantum phase fluctuation of laser ⁴³	117	117	1	~	N	N
Bi-phase State of OPO ⁴⁴	0.002	0.002	1	~	Y	N
Vacuum state fluctuation ⁴⁵	18.8	18.8	1	~2.3†	Y	Y
ASE ⁴⁶	16800	400	42	<40*	N	N
Laser diode (ASE) ⁴⁰	254000	2000	127	315*	Y	N
Chaotic laser ⁴⁷	320	320	1	16.7*	N	N
Chaotic laser ⁴⁸	1200	600	2	26*	N	N
Chaotic laser ⁴⁹	2240	320	7 (3)**	~	N	N
Chaotic laser ⁵⁰	10	10	1	11.6 †	Y	N

Chaotic laser ⁵⁴	21.1	21.1	1	7+	Y	N
Chaotic comb	300	20	15	5.6+	Y	Y
(this work)	3840	120	32	9.6+	Y	N

*Effective bandwidth, defined as the width of spectral segment that accounts for 80% of the total power.
†10-dB bandwidth.

**7 channels of random number generation, by linear combining 3 channels of optical chaotic signals.

Comment 2: “Despite the chaotic state studied in this work, there are various chaotic comb states such as the chaotic soliton state under anomalous dispersion and the chaotic comb state under normal dispersion. It is suggested to briefly comment on or compare the properties of those chaotic microcomb states under different dispersion regimes for random bit generation.”

Our reply: Yes, except for the chaotic comb state employed in this work, there are various chaotic state such as chaotic soliton state and chaotic comb state under normal dispersion. As for the chaotic soliton state, it is still a soliton state, whose generation process is quite challenging and complex. Also, the conversion efficiency is limited compared with chaotic comb state employed here.

Regarding the chaotic dark pulse state, since there is limited research on it, we provide a simulation result for the generation of a chaotic dark pulse in a microcavity under normal dispersion conditions, as depicted in Fig. R6. The simulation parameters are set as follows: the pump power is 100 mW, the intrinsic quality factor is 2×10^6 , the coupling quality factor is 2×10^6 , and the second order dispersion $D_2/2\pi$ is set to 1×10^6 . To obtain the dark pulse state, the resonant frequency of the pump mode is shifted by 1 GHz. Sweeping into the resonance, dark pulse state could be stimulated and a chaotic state could be obtained under proper detuning. The evolution of the optical spectrum and the pulse shape in time domain of the chaotic dark pulse state is recorded and shown in Fig. R6b and c. The average optical spectrum is shown in Fig. R6a. Compared with chaotic microcomb employed in this work, the spectrum is not flat at the center part. Additionally, as shown in Fig. R6d and e, the chaotic properties of different comb lines in the center part are quite varied, which is unfavorable for parallelization. Moreover, the inter-channel correlation is considerably high, even close to ± 1 , as shown in Fig. R7.

Thus, the chaotic comb used in this paper has advantages over other chaotic comb states in terms of system simplicity and consistency of performance among channels.

Fig. R6 Simulation results for the chaotic dark pulse. (a) the optical spectrum of the chaotic dark pulse. (b) the evolution of the optical spectrum. (c) the power evolution of the dark pulse in time domain. (d) the autocorrelation function of different comb lines with mode number $u = -1, -10, -20, -30$ and -40 . (e) the radio frequency spectra of different comb lines.

Fig. R7 Simulated inter-channel correlation between different comb lines.

Comment 3: “The chaotic signal detection employing the silicon photonics WDM receiver was

presented. While compared with the chaotic signal detected by commercial PD, the detected signal degenerated (Fig. 4). The Supplementary note IX only shows the autocorrelation function of detected signals under different bias voltages. It is still not clear what degenerates the signal. Was the signal filtered or distorted due to the bandwidth or the saturation power? Why the channel numbers are different in Fig.4 f, g?"

Our reply:

The degeneration is mainly caused by the inferior frequency response of the integrated Ge-Si photodiode compared with the commercial InP photodiode, as illustrated in revised Supplementary note X:

{Supplementary note X: Experiment: Comparison of commercial InP photodetectors and Si-Ge photodetectors}

Fig. S12. Comparison of integrated Ge-Si photodiode and commercial InP photodiode. a, the measured frequency response of the Ge-Si photodiode and InP photodiode. b, the frequency spectra of signals detected by Ge-Si photodiode and InP photodiode. c, Autocorrelation functions of chaotic signals detected by Ge-Si photodetectors and commercial InP photodetectors. d, the NIST SP 800-22 test results for random bits generated by InP photodiode with 30 Gbps generation rate.

In experiment, the chaotic signal is detected by commercial InP photodetectors and Si-Ge photodetectors respectively. In this part, we show that the frequency response or bandwidth of the Ge-Si photodiode is worse than that of the commercial photodiode, which degenerates the recorded signal. Fig. S12a shows the measured frequency response of the integrated Ge-Si photodiode and the InP photodiode. The Ge-Si photodiode shows a faster decrease at the high frequency. This is revealed at the frequency spectra of recorded signal as shown in Fig. S12b. Fig. S12c shows the autocorrelation function of the signal of the same comb line detected by the Ge-Si photodetector and the commercial InP photodetector employed in the main part. At different reverse biases, the detected signal express different features due to the relationship of the bias voltage and the bandwidth. The FWHM detected by the InP photodetector is 0.22 ns. While the FWHM detected by the Ge-Si photodetector is about 0.30 ns, due to the inferior frequency response. Employing the data recorded by the InP photodiode, the generation rate could reach 30 Gbps and the generated random bit sequences can pass the NIST SP 800-22 test successfully.

The channel number detected by the SiPh chip is mainly limited by the channel number of the wavelength division multiplexer. In our experiment, there are 30 comb lines within the gain band of the C-band EDFA and at the short-wavelength side of the pumped mode. Thus, in the detection employing discrete equipment, 30 channels are collected and applied for random sequence extraction, as shown in Fig. S12d. While for the integrated silicon photonics receiver, the channel space of the wavelength division multiplexer is about 180 GHz (as shown in Supplement Information IX), which is twice of the comb line space of our chaotic combs (~90 GHz). Thus, 15 channels are detected in experiment employing the integrated WDM receiver.

Comment 4: “Considering the random number extraction from a chaotic or random process, a flatter probability distribution tends to be more favorable. Previous research has shown that the intensity probability distribution of the chaotic comb line could be fitted by the Rice distribution, where the distribution function shows a rapid peak. This is also shown in Fig. 3e. Is it possible to change the probability distribution?”

Our reply: It is true that a flatter probability distribution is favorable for random bit generation. For the study that employing chaotic lasers for random bit generation, one of the aims is to flatten the probability distribution. Inspired by previous studies of chaotic lasers, in the revised manuscript, we employ two chaotic combs to rise the throughput rate and flatten the probability distribution. As shown in Fig. R8, by beating two comb lines from different chaotic combs, we could get a flatter probability distribution. As for the voltage normalized by the average, the standard deviations of the data for detecting one comb line and beating two comb lines are 0.698 and 0.812 respectively.

Fig. R8 The distributions of raw data employing one (a) and two (b) chaotic combs. the radio-frequency spectra of raw data employing one (c) and two (d) chaotic combs.

Fig. R8 also shows the spectra of one comb line and the beat signal of two comb lines. By beating two comb lines, we could get a wider and flatter radio frequency spectrum, contributing to a flatter probability distribution.

Comment 5: “Given the fact that the random bit stream relates to positive intensity values, the

correlation data presented in Fig.3h should be scaled in the range from 0 to 1. This would help understand the cross-correlation between off-diagonal channels.”

Our reply: Thanks for your suggestion. Considering a correlation calculated by

$$XCF_{m,n}(\tau) = \langle I_m(t + \tau) \cdot I_n(t) \rangle_t$$

the correlation should be a positive number as $I_m(t)$ is nonnegative. $I_m(t)$ is the recorded data, τ is the delay time, and $XCF_{m,n}$ is the correlation between $I_m(t)$ and $I_n(t)$. While in Fig. 2h (the original Fig. 3h), the cross-correlation is calculated by

$$XCF_{m,n}(\tau) = \frac{\langle \delta I_m(t + \tau) \cdot \delta I_n(t) \rangle_t}{\sqrt{\langle \delta I_m^2(t) \cdot \delta I_n^2(t) \rangle_t}}$$

Where $\delta I_m(t) = I_m(t) - \langle I_m(t) \rangle_t$, which is not nonnegative. Thus, the correlation shown in Fig. 2h could be negative. And some values of correlation between comb lines around the pump mode are negative as shown in Supplementary Fig. 5a. Thus, in the revised manuscript, log-scale is applied to show the inter-channel correlation:

Comment 6: “It is worth noting that some works (Iwami R, Mihana T, Kanno K, et al. Controlling chaotic itinerancy in laser dynamics for reinforcement learning[J]. Science Advances. 8(49), 2022) of optical decision using chaotic lasers show shorter convergence times even for problems with larger scales. Is the chaotic comb limit the convergence speed?”

Our reply: As for the scale, the number of slot machines were limited by the channel number of the

oscilloscope. In the original manuscript, we recorded four channels simultaneously. Thus, the experiments for $2^4=16$ slot machines were carried out. As for the convergence time, this was limited by the decision method or the algorithm. In the revised manuscript, we adopt another decision method, where one channel is correlated with one slot machines, expressing a higher convergence speed. Also, 32 channels at the short-wavelength side of the pump mode are recorded one-by-one, and 32 channels are employed for the MBA problem:

(Main part, page 6, line 22) Optical chaos, with its fast and complex internal time evolution, is a powerful entropy source that can be used for exploration purposes [54]. Chaotic lasers have been successfully employed to solve the multi-armed bandit problem (MAB) [29,55,56], a fundamental problem of reinforcement learning. To scale up the problem exponentially, a parallel scheme is required that employs parallel chaotic signals or is based on time-division-multiplexing [57]. In this section, we propose the use of the chaotic comb as a massively parallel chaotic source to solve the MAB. Fig. 4a shows the schematic of the optical decision making. There are N slot machines with a reward possibility $p_i (i = 0 \sim N)$ respectively. The decision maker needs to find out the slot machine with the highest reward possibility by consecutively playing the slot machines. For each play, the decision maker chooses and plays one slot machine. The decision maker will change its selection strategy based on whether a reward is obtained. In the decision maker, N chaotic signals $\Delta_i(t) (i = 0, 1, 2, 3)$ are detected and add with bias values $B_i(t)$ respectively to produce $A_i(t)$. Each channel is correlated with one slot machine. For each play, the slot machine with the highest biased signal $A_i(t)$ is selected and played. As continuously playing the game, bias values will be tuned based on tug-of-war method [56] (see Methods).

In experiment, we detect 32 channels at the short wavelength side of the pump mode one by one, to solve the 32-armed bandit problem. The hit probabilities of the 32 slots are set as follows: $P_1 = 0.7, P_2 = 0.5, P_3 = 0.9, P_4 = 0.1, \dots, P_{2j-1} = 0.7$ and $P_{2j} = 0.5$. As shown in Fig. 4b, the chaos-based decision maker would initially explore a wide range to identify the slot machine with the highest hit probability. After sufficient exploration, the slot machine 3 is identified as the best one and frequently selected, indicating successful decision making. To evaluate the performance of the decision making based on chaotic combs, the slot machines are played $C = 1000$ cycles consecutively, and this process is repeated $T = 1000$ times. For t -cycle, the correct decision ratio (CDR) is defined as T_{hit}/T , where T_{hit} represents the number of times a reward is obtained at the t -th cycle among the T processes. A CDR above 95% is viewed as a successful decision and the first cycle getting a CDR > 95% is defined as the convergence cycle (CC). Fig. 4c shows the CDR evolution along with the increase of cycles. Under the same system condition, the CDR evolution remains consistent across different N_{target} , which represent the number of the slot with the highest hit probability.

The scalability is a critical characteristic of the decision maker, where the convergence cycle will exponentially increase as the rise of problem scales. To assess the ability of solving large scale problems, we use the chaotic comb to solve the MBA problems with the number of slot machines $N = 4, 8, 16, 32, 64, 128, 256$. Limited by the gain bandwidth and the spectrum width of the chaotic comb, the number of available channels is 32. For $N > 32$, we record the 32 channels repeatedly to obtain enough channels of chaotic signals. As shown in Fig. 4d, the chaotic-comb-based decision marker can solve the MBA problems even with a large scale. Fig. 4e gives the performance comparison between chaotic-comb-based decision marker and other methods [56]. Fitted with the power function, the relationship between the convergence cycle CC and the number of slot machines N can be described as $CC = 26.48 \cdot N^{0.89}$. Compared with widely-used algorithm such as UCB1-tuned (upper confidence bound 1-

tuned) algorithm and Thompson sampling algorithm, methods based on optical-chaos show smaller scaling exponents, suitable for large-scale problems. Among the optical chaos, the decision maker shows a superb convergence speed, close to the spatial-chaos based method. While compared with the spatial chaos system, the chaotic microcomb benefits from integrated optoelectronics, providing a compact, low-cost, mass-producible decision marker. Despite having fewer channels compared to the spatial chaos system [56], the chaotic comb feathers high generation rate (40 GSa/s employed here for decision making) and the adoption of time-multiplexing [57] can be a promising approach to scale up the system, even though it may result in a tradeoff between speed and scalability.

Fig. 4 Multi-armed bandit problem solving based on chaotic combs. a, The scheme of the optical decision making based on the parallel chaotic source. b, One decision process for 32-armed bandit problem. The left figure shows the hit probability distribution of 32 slots, where the 3rd slot have the highest hit probability 0.9. c, the evolution of corrected decision rate with the increase of cycles. The red dashed line marks the corrected decision rate of 95%. d, The evolution of corrected decision rate under different scales. e, the comparison of scalability between chaotic-comb-based decision maker and other methods.

Comment 7: “The claim “...platforms, helps to achieve the full width at half maximum (FWHM) of all comb lines smaller than 0.15 ns... indicating remarkable stability and inter-channel consistency’ is not clear. Necessary explanations or references need to be added.”

Our reply: Thanks a lot for the reminder. A comparison in simulation of chaotic combs in different integrated optical nonlinear platforms is given in [Nature Photonics, DOI:10.1038/s41566-023-01158-4 (2023)]. While it is given by the ranging resolution. In Fig. R9, we show the FWHM of the auto-correlation function under different pump powers in different platforms.

Fig. R9 the FWHM of the auto-correlation function under different pump powers in different platforms.

Despite the high nonlinear, the strong thermo-optic effect makes the chaotic state could be stabilized under a larger detuning, making the chaotic comb able to maintain stable operation. This property is utilized for the stable operation of dark pulse microcomb in AlGaAsOI as we reported in [Nature, 605, 457 (2022)].

Correspond references are added in the revised manuscript:

(Main part, page 4, line 33) It is also worth noting that the strong Kerr effect in an AlGaAsOI microresonator, which holds the highest third-order nonlinear coefficient among all integrated nonlinear platforms [36,37], helps to achieve the full width at half maximum (FWHM) of all comb lines smaller than 0.15 ns. Fig. 2g shows the FWHM of the ACF changes with the detuning. Due to the self-locking induced by the intracavity thermal effect [38,39], the FWHM for all comb lines can be maintained within 0.2 ns for a detuning range larger than 25 GHz, indicating remarkable stability and inter-channel consistency.

Comment 8: “What is the pump power conversion efficiency, in comparison with other integrated nonlinear platforms? Is the chaotic comb line power sufficient for the photodetection considering the future integration with on-chip PDs? Will the additional on-chip optical amplification corrupt the randomness or the distribution, instead of using off-chip EDFAs?”

Our reply: At the present step, the conversion efficiency is 0.44% (26 dBm before injection into the chip and 2.4 dBm comb power with pump mode being suppressed), lower than chaotic combs generated in SiN microrings reported in [https://arxiv.org/abs/2112.10241] and [Optics express, 27, 19896 (2019)]. The main reason for the relatively low conversion efficiency is the critical coupling state and the existence of free carrier absorption induced by the three-photon absorption. An over-coupling microring would be favorable for the enhancement of conversion efficiency. Additionally, if the FCA could be weakened by introducing the PIN structure to shorten the effective carrier life time, we could get a conversion efficiency up to 70.33% as shown in Supplementary Fig. 7c. Also, as shown in Supplementary Fig. 7d, a chaotic comb with -5 dBm per line is possible which is high enough for integrated Ge-Si photodetectors.

Considering the present step, where the coupling state and the strong FCA blocks a high conversion efficiency. Optical amplifier should be employed. In the revised Supplementary Information, we add a part about the comparison between EDFA and SOA:

{Supplementary note XII: The influence of different amplification processes}

Fig. S14 The influence of different amplification processes. a, the schematic of the test link. b, the optical spectrum before amplification. c, the optical spectra after the first order amplification. d, the optical spectrum after the first order amplification around the C-band. The spectrum of SOA is filtered by a band pass filter. e, the optical spectrum after the second order amplification. f, the radio-frequency spectra of the recorded comb line. g, the autocorrelation functions of the recorded comb line.

In the experiment described in the main text, we used a commercial EDFA to amplify the generated chaotic comb due to its weak comb line power. Here, we compare the amplification performance of commercial EDFAs and semiconductor optical amplifiers (SOAs). Supplementary Fig. 14 shows the experimental setup. Unlike the main text, where an optical amplifier was arranged between the bandpass filter and the photodiode, an electrical amplifier is arranged after the photodiode to amplify the detected signal and compensate for the limited response of the commercial InP photodiode. First, we record the original optical spectrum of the chaotic comb, as shown in Supplementary Fig. 14b. The signal-tonoise ratio (SNR) of comb lines within the C-band can reach 40 dB. Supplementary Fig. 14c shows the spectra of amplified combs. The first order amplification of the EDFA is set to the maximum. Different from the EDFA, the gain bandwidth of the SOA is wider. To obtain higher power in the C-band, a bandpass filter was employed after the first SOA to suppress comb lines outside the C-band. The filtered spectrum is shown in Supplementary Fig. 14d. Due to a higher noise figure compared to the EDFA, the comb lines amplified by the SOA show lower SNR. For the comb line around 1545 nm, the SNR is 23.7 dB, which is 4.3 dB lower than that amplified by the EDFA. Moreover, due to the insertion loss of the bandpass filter (5 dB), the power amplified by the SOA is lower. Thus, the filtered comb is amplified by another SOA. Supplementary Fig. 14e shows the optical spectrum after the second SOA. In addition, for the EDFA, the second-order amplifier is turned on and set to the maximum. The output powers of the SOA and the EDFA are 16 dBm and 22 dBm, respectively. Thus, the comb line amplified by the SOA is weaker. One comb line around 1545 nm is filtered out by a tunable bandpass filter with 6

dB insertion loss. The comb line power sent to the photodiode is -3 dBm and -8 dBm for the EDFA and SOA, respectively. Supplementary Fig. 14f shows the radio frequency spectra amplified by the EDFA and SOA. Due to a higher noise figure and lower saturation power, the signal amplified by the SOA shows a 3.1 dB lower SNR. Supplementary Fig. 14g shows the auto-correlation function of the comb line amplified by the SOA and EDFA respectively. The two lines are almost coincident with each other. Thus, at the present set up, the random from the chaotic comb is not corrupted by the replacement of EDFA with SOA.

As shown in the above, the SNR using SOAs is about 3 dB lower than that employing commercial EDFA. This do not corrupt the recorded signal. 30 Gbps random bit generation can be realized and pass the NIST SP 800-22 test successfully using the signal amplified by SOAs.

Comment 9: “The authors used differentiation to artificially increase the sampling rate, make the distribution symmetric, and reduce the correlation, which is a widely used approach. The authors should explicitly introduce the idea of this approach with references.”

Our reply: Thanks for your suggestion. The high order derivative is a powerful post-processing method for fast random bit generation by using slow signals, as shown in Fig. R10 from [Physical review letters, 111, 044102 (2013)].

Fig. R10 the schematic of the high order derivative

It could artificially increase the sampling rate to realize an ultra-high output rate with complex post-processing. While in this work, we applied only one step of self-delayed differentiation. The main propose of using self-delayed differentiation is for unbiased bits. To show the influence of self-delayed differentiation, we gives the extracted 3 LSBs with (Fig. R11a) and without (Fig. R11b) differentiation.

Fig. R11 the distribution of extracted 3 LSBs with (a) and without (b) differentiation

As shown in Fig. R11b, without the differentiation, the distribution of extracted 3 LSBs is not flat. Some bit patterns (000, 010, 011, 100) appear more frequently compared with the others. This leads to a

“biased” bit sequence.

To make it clearer, the description in the manuscript is revised as following:

(Main part, page 4, line 102) As shown in Fig. 2e, the intensity distribution of the output I shows an asymmetric shape, ~~which leads to biased bits, therefore degrading the quality of random bit generation.~~

To symmetrize the distribution, a delay difference is employed to the raw data, obtaining ΔI with a symmetric distribution as shown in Fig. 3c. ~~The symmetric distribution is favorable for unbiased bit extraction [41].~~

Response to the report from the Referee #3

Comments: *“The manuscript written by B. Shen and co-authors presents the implementation of the random number generator based on the chaotic state of the microcomb. In this work, a highly nonlinear AlGaAsOI resonator with ~100 GHz free spectral range has been used for this purpose. The chaos-assisted random number generation has been applied to the multi-armed bandit (MAB) problem.*

Despite the excellent presentation and quality of the presented data, the novelty of the manuscript is rather limited:”

Our reply: We thank the referee for carefully reading our manuscript and the evaluation of our work. We made the point-by-point response according to your comments as follows.

Comment 1: *“The chaotic state of the nonlinear resonator is rather a developed topic. Indeed, chaotic states of the microresonator have been achieved and demonstrated approximately 15 years ago. The very first demonstrated microcomb was found in this particular state. This state can be achieved even in the devices having large internal losses, mode crossings, and so, which would be an obstacle for the generation of coherent dissipative solitons. To my point of view all the results presented until section “Integrated parallel random bit generation” are almost completely covered by the three following papers [Coulibaly, et al. PRX 2019 {10.1103/PhysRevX.9.011054}, Wang, et al. Optics Express 2012 {10.1364/OE.20.029284}, Erkintalo, et al. Optics Letters 2014 {10.1364/OL.39.000283 }] as well as in Ref [11] of the manuscript.”*

Our reply: We agree with the referee that the chaotic microcomb is a developed topic. This paper's significant contribution is that **we propose and verify, for the first time, the feasibility of using chaotic optical combs as parallel chaos sources.** There are two critical issues that need to be addressed in using chaotic optical combs as parallel chaos sources: **the chaotic performance of each tooth of the chaotic optical comb and the correlation between each tooth of the chaotic optical comb.**

1. The chaotic performance: for most previous works such as [Physical Review X, 9, 011054 (2019)], the whole chaotic comb, while not one comb line of the chaotic comb, was studied. In ref [11], the bandwidth and full-width-half-max of autocorrelation of each comb line were valued, while the inter-channel correlation was not studied. In [Nature Communications, 12, 427 (2021)], the correlation between only two certain channels were valued. In this paper, we conduct a **thorough characterization of the chaotic optical comb.** Through experiments, we evaluate the autocorrelation characteristics of

all comb teeth within the C band. The comb teeth near the pump mode exhibit similar chaotic performance. Additionally, we observe that the power of each comb tooth is similar. It is worth noting that due to **the excellent nonlinear effect of AlGaAs**, in this paper we can obtain **superior chaotic optical combs with moderate pump power**. Compared with the radio-frequency spectrum width of all integrated chaotic optical combs reported in the past, the chaos bandwidth generated by using AlGaAs ring is the widest. The performance of our current optical comb is limited by the carrier absorption effect. If we can reduce carrier lifetime and weaken the effect of carrier absorption by introducing structures such as PIN, we are expected to achieve a chaotic optical comb with higher performance. In addition, AlGaAsOI microrings heterogeneously integrated with the SOI platform have been reported in [Photonics Research, 10, 02000535 (2022)], and the corresponding quality factor is up to 1.12×10^6 , which is sufficient to support the excitation of chaotic optical combs. By combining high power on-chip laser, high nonlinear microring and silicon photonics, it is promising to realize **fully integrated parallel chaotic source generation and processing chip as illustrated in Fig. 1**.

2. The inter-channel correlation: in this work, **we give the inter-channel correlation between each pair of comb lines for the first time, both in experiment and in simulation**. While intensity noise correlation was presented in a previous study [Nature Communications, 12, 427 (2021)], it is worth noting that this only evaluated the correlation between two certain channels. For spatiotemporal or multi-output chaotic systems, the general inter-channel correlation cannot be easily verified by calculating the correlation between certain channels. For example, periodic inter-channel correlation was observed in [Physical review letters, 77, 4162-4165 (1996)]. Therefore, **it remains unclear whether the chaotic signal carried by each comb line of chaotic combs is correlated with other lines**. Our work fills this gap by providing a full investigation of inter-channel correlation between each pair of comb lines, and two properties are revealed. First, **for comb line pairs distributed symmetrically on both sides of the pump mode, positive correlation exists**, which was not observed in [Nature Communications, 12, 427 (2021)], due to the lack of a full investigation of every comb line pairs. Second, for other comb line pairs around the pump mode, **the inter-channel correlation can be lower than 0.04**. Our finding suggests that it would be a proper choice to use comb lines on the same side of the pump mode as parallel chaotic sources. Low inter-channel correlation is essential for applications ranging from private communication and key distribution to chaotic lidar and radar. In the private communication and key distribution, low inter-channel correlation ensures the security and confidentiality of the transmitted information by generating a random key that cannot be easily intercepted or decoded [Physical review letters, 77, 4162 (1996)]. In chaotic lidar/radar systems, low inter-channel correlation helps to reduce interference and improve the system's accuracy by increasing the signal-to-noise ratio [Nature Photonics, DOI:10.1038/s41566-023-01158-4 (2023)].

Comment 2: "Random number generator operating in the quantum regime has been implemented with integrated microresonators earlier with a different scheme [Okawachi, et al. Optics Letters 2016]."

Our reply: Employing the kerr nonlinearity in the microcavity, the bi-phase state of optical parametric oscillation could produce quantum random number. While the microring is the only integrated devices in the random bit generation system.

Compared with [Okawachi, et al. Optics Letters 2016], our work is different in the following: 1) In this

work, the random bit generator proposed in this work employs chaotic combs as massively parallel chaotic sources to extract random bits. It is **different in principle**. 2) Considering the generation rate, our method achieves a **high throughput rate** of 10s Gbps, meeting the requirements of high-speed information systems. Our revised manuscript details the comparison between our method and previous methods, as discussed in "**reply to comment 6**". By combining two chaotic combs, we can achieve 120 Gbps random bit generation pre-channel. 3) The adoption of chaotic combs provides **massive parallelism** for the random bit generation. In this work 15-channel and 32-channel parallel random bit generation are demonstrated with integrated photodiodes and commercial InP photodiodes. 4) In this work, we give the first demonstration of parallel chaotic **signal detection in integrated platforms**.

Comment 3: "3) MAB problem parallelization has already been demonstrated in photonics for the 512 arm version [44]."

Our reply: Yes, the previous work [Optica, 10, 339 (2023)] gave an excellent result for large scale MBA problems. In the revised manuscript, we modified our experiment for MBA problem in whole, employing the same algorithm in [Optica, 10, 339 (2023)]. It shows a comparable scaling exponent with the spatial-chaos scheme. Despite weaker scalability of frequency domain (1-dimension space) compared with spatial domain (2-dimension space), chaotic-comb-based scheme benefits from the integrated optoelectronics, providing a compact, low-cost, mass-producible decision marker. Also, the time-division-multiplexing could be applied to enlarge the scale, thanks to a faster variation of chaotic combs compared with spatial chaos system.

More detailed description and comparison can be found in "**reply to comment 7**".

Comment 4: "In this regard, the title "Massively parallel chaotic sources based on microcombs" is too general and must be modified."

Our reply: Thank you for your feedback. We appreciate your suggestion for improving the title of our manuscript. After careful consideration, we have revised the title to "Harnessing microcomb-based parallel chaos for integrated photonics" to better reflect the focus of our research. We hope this revised title accurately represents our work and addresses your concern.

Comment 5: "However, the combination of the tool, method, and problem is indeed novel and to my point of view can be considered for publication, if parts that represented the interest for the community are restructured and straightened.

The original part of the work is dedicated to the experimental realization of the random bit generator applied to the MAB demonstration seems to be interesting in the context of the emerging applications of the microresonator chaos. However, a detailed comparison with the existing methods must be improved and extended. As an example, the following questions can be answered:"

Our reply: We thank the referee for positive comments on the originality of our work. We agree with the referee that the previous manuscript did not highlight the novelty. Therefore, we have restructured and revised the manuscript to emphasize the novelty of our work. The main changes that we have made

are outlined below:

1) We have moved the first part of the Result section, *Route into the chaotic microcomb*, to Supplementary note III, and added a review and comparison of previous works about chaotic combs in the Introduction part.

2) A comparison of different methods for optical chaos and random bit generation based on optical sources is given in Table 1.

3) We have modified the experiment in the last part of the Result section, *Computation acceleration based on chaotic combs*, to show a fast convergence speed that is suitable for large-scale problems.

You can find a detailed description of each revision in the corresponding sections below.

Comment 6: “Where is the place of the chaotic microcombs in the list of state-of-the-art methods according to the power budget, bandwidth, and autocorrelation? Several methods to generate chaotic optical signals must be compared.”

Our reply: Thanks for your suggestion. **The chaotic comb is a promising scheme for fully integrated parallel chaotic source.** One widely-used scheme for optical chaos is based on chaotic lasers, where a specific feedback loop with a laser diode is used for per channel. In order to obtain parallel output, multiple semiconductor lasers are needed. Compared with optical chaotic combs, the cost for integration and control of laser array is much higher, similar to the comparison between the soliton microcomb and the laser array for optical communication, especially when considering the need for signal transmission using wavelength-division multiplexing technology or chaotic lidar using dispersion elements. In terms of single-channel performance, we must acknowledge that the current single-channel performance of chaotic optical combs is inferior to that of chaotic lasers in terms of single-channel power and chaotic bandwidth. However, as discussed in our text and Supplementary note VIII, by introducing PIN structures to reduce carrier lifetime and carrier absorption, single-channel performance comparable with chaotic laser performance can be obtained. Additionally, the power of each comb tooth can be improved by designing the coupling coefficient to improve the conversion efficiency of the optical comb. In this paper, to compensate for the disadvantage of chaotic bandwidth, we also attempt to use the scheme of combining two optical combs, inspired by the scheme of chaotic lasers [Optics Express, 28, 3686 (2020)]. By combining multiple chaotic combs, we are able to increase radio frequency bandwidth and achieve 120 Gbps random bit generation in a single channel. The other scheme is based on spatial-temporal chaos in space optics, which has excellent applications in reserve pool calculation and other fields. Based on spatial dimension, spatial optical chaos can realize large-scale chaotic signal distribution. However, it should be noted that this scheme is difficult to integrate because it is based on spatial optics. Contributed to the massively parallelism and the feasibility for integration of chaotic microcombs, the total bit rate based on chaotic microcombs is the highest among optical-chaos scheme.

In the revised manuscript, we have added a comparison of optical chaotic sources in Table. 1:

(Main part, page 5, line 7) **The generation rate of single chaotic comb is mainly limited by the chaotic bandwidth, which is degenerated by the nonlinear absorption. To compensate this, here we provide a method by a dual-comb scheme. The schematic is shown in Fig. 3c, where two combs are pumped by**

two lasers with a frequency difference around 4 GHz. In experiment, two microcavities are packaged with temperature controllers and the chaotic combs are generated by sweeping the resonant frequency by tuning the chip temperature, while the pump lasers are kept fixed. The comb teeth of the two chaotic combs are filtered, amplified and combined, before being sent to the photodiode. The time domain waveforms of each tooth pairs are recorded at a sampling rate of 80 GSa/s. The raw data are down-sampling to 40 GSa/s and processed as described above. In this case, the single channel generation rate can be increased to 120 Gbps, which is comparable to the novel random bit generator based on chaotic lasers. 32 channels on the same side of the pump mode are recorded, processed, and successfully pass the NIST SP 800-22 test (see in Fig. 3h), corresponding to an aggregation rate of 3.84 Tbps.

Fig. 3. The parallel random bit generator based on a microresonator and a SOI chip. a, Optical microscope photograph of the SiPh receiver. b, The set-up scheme for parallel random number generation; AWG, arrayed waveguide grating; PD, photodetector; OSC, oscillator; D, delay unit; XOR, exclusive-OR. c, The setup for random bit generation using two chaotic combs. d, The possibility distribution function of the differential data. e, The distribution of the extracted 3 LSBs. f, The ACF of the generated bit sequence. The red line indicates the lower limit determined by $1/\sqrt{n}$. The NIST SP800-22 test results for signals detected by SOI PD with setup shown in b (g) and commercial InP PD with setup shown in c (h).

In Table I, we present a comparison of different random bit generation schemes. Due to the massively parallelism provided by chaotic combs, the total bit rate of chaotic-comb-based RBG shows the highest generation rate among optical-chaos-based methods. As the lack of data in published articles, the power consumption is not listed in Table I. Considering simple setup with one pump laser and one microcavity, the generation of optical chaotic sources based on chaotic combs is power efficient and

low-cost compared with chaotic lasers [52,53], which require at least one laser diode per channel. In addition, our work gives the first demonstration of parallel chaotic signal detection employing the integrated SiPh chip. Although the generation rate of chaotic-comb based RBG is lower than ASE (amplified spontaneous emission)-based RBG with spectral or spatial parallelism, the use of chaotic systems offers the ability to synchronize the output of two chaos generators [4,35], which can be arranged at the transmitter and receiver sides respectively. This synchronization enables key distribution and private communication, which is not possible with stochastic sources such as ASE and random lasers.

Table 1. Comparison of different schemes of random bit generation based on optics

Scheme	Total bit rate (Gbps)	One channel (Gbps)	Channel number	Bandwidth (GHz)	Integrated source	Integrated processor
Random laser ⁴²	540	540	1	29 †	N	N
Quantum phase fluctuation of laser ⁴³	117	117	1	~	N	N
Bi-phase State of OPO ⁴⁴	0.002	0.002	1	~	Y	N
Vacuum state fluctuation ⁴⁵	18.8	18.8	1	~2.3†	Y	Y
ASE ⁴⁶	16800	400	42	<40*	N	N
Laser diode (ASE) ⁴⁰	254000	2000	127	315*	Y	N
Chaotic laser ⁴⁷	320	320	1	16.7*	N	N
Chaotic laser ⁴⁸	1200	600	2	26*	N	N
Chaotic laser ⁴⁹	2240	320	7 (3)**	~	N	N
Chaotic laser ⁵⁰	10	10	1	11.6 †	Y	N
Chaotic laser ⁵⁴	21.1	21.1	1	7†	Y	N
Chaotic comb	300	20	15	5.6†	Y	Y
(this work)	3840	120	32	9.6†	Y	N

*Effective bandwidth, defined as the width of spectral segment that accounts for 80% of the total power.

†10-dB bandwidth.

**7 channels of random number generation, by linear combining 3 channels of optical chaotic signals.

Comment 7: "In this regard, can the chaotic microcomb be considered as a competitive source for the random number generation?"

Our reply: As discussed in the revised manuscript (as shown in "reply to comment 6"), the chaotic-comb-based scheme is competitive among optical-chaos-based schemes, with the highest total bit rate. In addition, in this work we give the first demonstration of parallel chaotic signal detection on integrated platforms employing the silicon photonics. The AlGaAsOI platform is fabricated by heterogeneous integration on Si substrate, and thus is compatible with the most widely used platform of silicon photonics. Such integration has recently been demonstrated, suggesting that our chaotic microcomb can be seamlessly implemented in fully integrated photonic systems, together with diverse

silicon photonic engines. As demonstrated in Supplementary note VI and X, it is feasible to construct a fully integrated parallel chaotic signal generation, detection, and processing system without the need for an optical amplifier.

Also shown in Table. 1, the generation rate of chaotic-comb-based RNGs is lower than that of stochastic sources such as ASE. While the use of chaotic systems allows for the synchronization of the output of two chaos generators, which can be placed at the transmitter and receiver sides, respectively. This synchronization enables key distribution and private communication, which is not possible with stochastic sources such as ASE and random lasers.

Comment 8: “Why does the MAB problems has been limited to 16 slots in the current work? What is the estimated capacity to scale this system?”

Our reply: In the previous manuscript, the scale of 16 slots was limited by the number of channels recorded simultaneously by the oscilloscope, which is 4. Thus, the scale was $2^4=16$. In the revised manuscript, we recorded 32 channels one by one of the same chaotic comb. If adopting the decision method presented in previous manuscript, the scale could reach 2^{32} . However, using the previous method, the convergence speed would decrease rapidly as the problem scale increases. Therefore, we have adopted a different method in the revised manuscript where one chaotic channel is connected to one slot machine:

(Main part, page 6, line 22) Optical chaos, with its fast and complex internal time evolution, is a powerful entropy source that can be used for exploration purposes [54]. Chaotic lasers have been successfully employed to solve the multi-armed bandit problem (MAB) [29,55,56], a fundamental problem of reinforcement learning. To scale up the problem exponentially, a parallel scheme is required that employs parallel chaotic signals or is based on time-division-multiplexing [57]. In this section, we propose the use of the chaotic comb as a massively parallel chaotic source to solve the MAB. Fig. 4a shows the schematic of the optical decision making. There are N slot machines with a reward possibility $p_i (i = 0 \sim N)$ respectively. The decision maker needs to find out the slot machine with the highest reward possibility by consecutively playing the slot machines. For each play, the decision maker chooses and plays one slot machine. The decision maker will change its selection strategy based on whether a reward is obtained. In the decision maker, N chaotic signals $\Delta_i(t) (i = 0, 1, 2, 3)$ are detected and add with bias values $B_i(t)$ respectively to produce $A_i(t)$. Each channel is correlated with one slot machine. For each play, the slot machine with the highest biased signal $A_i(t)$ is selected and played. As continuously playing the game, bias values will be tuned based on tug-of-war method [56] (see Methods).

In experiment, we detect 32 channels at the short wavelength side of the pump mode one by one, to solve the 32-armed bandit problem. The hit probabilities of the 32 slots are set as follows: $P_1 = 0.7, P_2 = 0.5, P_3 = 0.9, P_4 = 0.1, \dots, P_{2j-1} = 0.7$ and $P_{2j} = 0.5$. As shown in Fig. 4b, the chaos-based decision maker would initially explore a wide range to identify the slot machine with the highest hit probability. After sufficient exploration, the slot machine 3 is identified as the best one and frequently selected, indicating successful decision making. To evaluate the performance of the decision making based on chaotic combs, the slot machines are played $C = 1000$ cycles consecutively, and this process is repeated $T = 1000$ times. For t -cycle, the correct decision ratio (CDR) is defined as T_{hit}/T , where T_{hit} represents the number of times a reward is obtained at the t -th cycle among the T processes. A CDR above 95% is viewed as a successful decision and the first cycle getting a CDR > 95% is defined as the convergence cycle (CC). Fig.

4c shows the CDR evolution along with the increase of cycles. Under the same system condition, the CDR evolution remains consistent across different N_{target} , which represent the number of the slot with the highest hit probability.

The scalability is a critical characteristic of the decision maker, where the convergence cycle will exponentially increase as the rise of problem scales. To assess the ability of solving large scale problems, we use the chaotic comb to solve the MBA problems with the number of slot machines $N = 4, 8, 16, 32, 64, 128, 256$. Limited by the gain bandwidth and the spectrum width of the chaotic comb, the number of available channels is 32. For $N > 32$, we record the 32 channels repeatedly to obtain enough channels of chaotic signals. As shown in Fig. 4d, the chaotic-comb-based decision marker can solve the MBA problems even with a large scale. Fig. 4e gives the performance comparison between chaotic-comb-based decision marker and other methods [56]. Fitted with the power function, the relationship between the convergence cycle CC and the number of slot machines N can be described as $CC = 26.48 \cdot N^{0.89}$. Compared with widely-used algorithm such as UCB1-tuned (upper confidence bound 1-tuned) algorithm and Thompson sampling algorithm, methods based on optical-chaos show smaller scaling exponents, suitable for large-scale problems. Among the optical chaos, the decision maker shows a superb convergence speed, close to the spatial-chaos based method. While compared with the spatial chaos system, the chaotic microcomb benefits from integrated optoelectronics, providing a compact, low-cost, mass-producible decision marker. Despite having fewer channels compared to the spatial chaos system [56], the chaotic comb feathers high generation rate (40 GSa/s employed here for decision making) and the adoption of time-multiplexing [57] can be a promising approach to scale up the system, even though it may result in a tradeoff between speed and scalability.

Fig. 4 Multi-armed bandit problem solving based on chaotic combs. a, The scheme of the optical decision making based on the parallel chaotic source. b, One decision process for 32-armed bandit problem. The left figure shows the hit probability distribution of 32 slots, where the 3rd slot have the highest hit probability 0.9. c, the evolution of corrected decision rate with the increase of cycles. The red dashed line marks the corrected decision rate of 95%. d, The evolution of corrected decision rate under different scales. e, the comparison of scalability between chaotic-comb-based decision maker and other methods.

As discussed in the last paragraph in the revised manuscripts above, the scale of present chaotic comb is limited by the gain bandwidth and the spectrum width of the chaotic comb, where 32 parallel chaotic

comb lines are available. The time multiplexing technology can be employed to solve problems with larger scales at the cost of speeds. While compared with spatial-chaos-based method in [Optica, 10, 339 (2023)], where the speed will be limited by 60 frame/s refresh speed of spatial light modulator, the generation speed for parallel chaotic signals is much higher even employing the time division method. For example, solving the 512-armed bandit problem, each raw channel could be divided into 16 channels and the generation speed for divided channels is 2.5 GHz, which is much higher than 60 frame/s. In addition, as we have discussed in Supplementary note VI and X, it is feasible to obtain a chaotic signal generation and detection system without the need for optical amplifiers, potentially increasing the available channels to hundreds by carefully tuning the cavity dispersion. Combined with the small scaling exponent and the integrated optoelectronics, the chaotic comb provides a promising scheme for a massively parallel chaotic source for large-scale MBA problems.

Comment 9: “What makes this chaotic microcomb different in comparison to all the prior devices demonstrated during last 15 years?”

Our reply: As we have discussed in "**reply to comment 1**", the difference of our work compared with prior devices and works demonstrated during last 15 years is: **for the first time, we propose and verify the feasibility of chaotic optical combs as parallel chaos sources**. It has superior optical parallel chaos generation capability compared with other integrated method. Also, it possesses the broadest chaotic bandwidth currently known among all the microcomb platform due to highest nonlinear effect of AlGaAs. Moreover, we show **the first demonstration of the combination of integrated parallel chaotic sources and silicon photonics** for the chaotic signal generation and detection systems (see "**reply to comment 7**"). Our work paves the way for chaos-based information processing systems using integrated photonics.

To highlight the novelty, we take following actions:

1. Review and comparison of previous works about chaotic combs are added in the introduction part: (Main part, page 2, line 17) **The use of a chaotic comb as a parallel chaotic source offers a promising alternative, as spatiotemporal chaos can be achieved by pumping an optical nonlinear microcavity with sufficient power. In frequency domain, the comb is consisted of well-separated comb lines, each exhibiting poor coherence [31–33]. This promises tremendous parallelism provided by optics through wavelength division-multiplexing technology. Recently, chaotic combs have been employed as parallel optical sources for optical coherence tomography [34] and parallel ranging [11], but the parallelization was based on frequency domain rather than on the chaotic signal carried by each channel. For parallel chaotic sources, channels should be uncorrelated with each other, which is vital to applications ranging from private communication [4,35] and key distribution [5] to chaotic lidar/radar systems [7,36]. However, it is still unclear whether chaotic combs can function as parallel chaotic sources. In this work, we fill this gap by providing a full investigation of chaotic comb lines, and introduce the optical chaotic comb as a massively parallel chaotic source for photonic integrated circuits.**

2. The first part of the Results, *Route into the chaotic microcomb*, is moved to the Supplementary note III.

Comment 10: “Minor remark: Avoided mode crossings are very pronounced in the integrated dispersion profile as follows from Sup. Fig. 7. It would be beneficial for the reader to explore how the avoided mode crossings influence the properties of chaos by direct numerical simulations.”

Our reply: Thank you for your suggestion. In our revised Supplementary note VII, we have included the radio-frequency spectrum and autocorrelation function both with and without mode crossing:

Fig. R12 the auto-correlation function and radio frequency spectrum with (blue) and without (red) mode crossing.

In this simulation case, it is worth noting that the presence of mode crossing results in a wider RF spectrum. This wider spectrum may be attributed to the broader resonance and lower quality of the microcomb caused by the mode crossing. However, the impact of mode crossing on microcombs is a complex problem, which requires further investigation in future work.

Comment 11: “Concluding, I find that applications of the chaotic microcombs a promising direction of research with only a few existing works so far. However, the novelty of the presented data in this particular case is questionable. Thus, this manuscript can be re-considered for publication after major revision that must include removal of the data that does not have significant novelty, comparison of the device performance with existing photonic platforms, and clarification of the scalability of the system, ideally with experimental proof.”

Our reply: Thank you for your valuable feedback. We appreciate your positive comments on the potential of chaotic microcombs and agree that more research is needed in this direction. We have carefully considered your comments and revised our manuscript accordingly, as we have replied above. We hope that our revisions have addressed your concerns, and we look forward to your review of our revised manuscript.

REVIEWER COMMENTS

Reviewer #1 (Remarks to the Author):

I would like to thank the authors for carefully addressing my comments.

Please, find my comments to the replies below.

Comment 2: Please cite Ref by Marchand et al. It is a good scientific practice to acknowledge prior research in the field.

Comment 5: Comment on the relation between source bandwidth and achievable sampling rate.

Comment 4: Explicitly mention that optical chaos synchronization has not been demonstrated (including MI) or give relevant references. Ref 4 is a simulation, Ref 35 employed electrical circuits.

Comment 7: You could apply delay difference (few times, if necessary) to symmetrize the distribution (similar to Fig3 results) and repeat the analysis.

Reviewer #2 (Remarks to the Author):

In the revised manuscript, the authors have well addressed my concerns. In my opinion, this work does not aim to promote the scientific advance of chaotic microcomb physics but made substantial efforts to apply this emerging technology for practical applications such as RNG and scalable decision making. Thus, I think the successful achievement of interesting application demonstrations of integrated chaotic microcomb sources can surely meet the criteria of Nature Communications acceptance.

I can recommend this work for publication, but I still have a minor suggestion that the authors could highlight the work (already cited as Ref [11]) on parallel LiDAR sensing based on integrated chaotic microcombs, which is in fact one of the early applications of on-chip chaotic microcomb sources.

Reviewer #3 (Remarks to the Author):

The replies given by the authors are satisfactory. However, I must insist that the given work is by far not the first that proposes to harness microcomb-based chaos. Therefore, I find the title, and the general presentation of the article very misleading.

We appreciate the careful review by the reviewers and have modified the manuscript in accordance with their suggestions. Here, we present a point-by-point reply (in blue) to the reviewers' comments (in black), as well as the action taken (in red).

Response to the report from the Referee #1

Comments: *"I would like to thank the authors for carefully addressing my comments. Please, find my comments to the replies below."*

Our reply: We appreciate the reviewer for carefully reviewing our revised manuscript and for helpful comments. These comments, followed by our point-by-point responses, are shown below.

Comment 1: "Comment 2: Please cite Ref by Marchand et al. It is a good scientific practice to acknowledge prior research in the field."

Our reply: Thanks for the suggestion. In the revised manuscript, [Nature Communications, 12, 427 (2021)] was cited:

(Main part, page 2, line 11) Recently, chaotic combs have been employed as parallel optical sources for optical coherence tomography [34,35] and parallel ranging [11].

[35] Marchand, P. J. et al. Soliton microcomb based spectral domain optical coherence tomography. Nature Communications 12, 427 (2021).

Comment 2: "Comment 5: Comment on the relation between source bandwidth and achievable sampling rate."

Our reply: The relationship between the source bandwidth and achievable sampling rate is unclear. According to the Nyquist theorem, sampling at the twice of the bandwidth of signal is enough for harvesting entropy. But it is possible to sampling faster than the Nyquist rate with employing post-processing to obscure inter-sampling correlation. While this will not increase the rate of entropy production, which limits the random bit generation rate. Thus, we would like to give a comment on the relationship between the signal bandwidth and the random number generation rate. As we have shown in the response letter of the last round, the generation rate should be limited by the entropy of the raw signal. The entropy h_0 can be estimated by the following equation:

$$h_0 = \min(\tau^{-1}, 2BW)(N_\epsilon - D_{KL}(p(x)||u(x)))$$

Where τ is the sampling period, BW is the bandwidth of the entropy source, N_ϵ is the number of bits per sample, $p(x)$ is the probability density function of the entropy source, $u(x)$ is the uniform distribution and D_{KL} is the Kullback-Leibler divergence from $u(x)$ to $p(x)$. We can see that the entropy is limited by the twice of bandwidth and the probability density function. The entropy can be increased with a larger bandwidth and a more flat probability density function. Thus, we can get a higher generation rate by combining two chaotic microcombs. Corresponding comments are added in the revised Supplementary note 1.

Comment 3: "Comment 4: Explicitly mention that optical chaos synchronization has not been

demonstrated (including MI) or give relevant references. Ref 4 is a simulation, Ref 35 employed electrical circuits.”

Our reply: The synchronization of chaotic combs has not been demonstrated in experiment. While the synchronization of optical chaos has been well developed both in experiment and in simulation. [Physical Review Letters, 72, 2009 (1994)] gives the first demonstration of synchronization of chaotic lasers and the synchronization of chaotic lasers have been employed for private communication [Nature, 438, 343 (2005)] and key distribution [Light: Science & Applications, 10, 1 (2021)]. While the synchronization of chaotic combs has not been demonstrated both in experiment and in simulation as far as we know:

(Main part, page 6, line 7) Although the generation rate of chaotic-comb based RBG is lower than ASE (amplified spontaneous emission)-based RBG with spectral or spatial parallelism, the use of chaotic systems offers the ability to synchronize the output of two chaos generators, **which has been well studied for chaotic lasers [2,5] but not demonstrated yet for chaotic combs.** The synchronized systems can be arranged at the transmitter and receiver sides respectively.

[2] Argyris, A. et al. Chaos-based communications at high bit rates using commercial fiber-optic links. Nature 438,343–346 (2005).

[5] Gao, H. et al. 0.75 gbit/s high-speed classical key distribution with mode-shift keying chaos synchronization of fabry–perot lasers. Light: Science & Applications 10,1–9 (2021).

Comment 4: “Comment 7: You could apply delay difference (few times, if necessary) to symmetrize the distribution (similar to Fig3 results) and repeat the analysis.”

Our reply: The result shown in the last response letter were processed by one step of self-delayed difference. Here, two steps of self-delayed differentiation are applied to the sampled noise from the scope as shown in Fig. R1a.

Fig. R1 (a) The two-step self-delayed difference process. (b) The distribution of processed data. (c) The autocorrelation function of the processed data. (d) The distribution of extracted 3 LSB.

Fig. R1b and d show the symmetrized and flattened distributions of the processed data and extracted 3 LSB, respectively. However, despite the flattened distributions, the extracted bit sequence did not pass the NIST SP 800-22 test. This may be due to the internal periodicity of the recorded data, which is

induced by the internal clock signals. Fig. R1c shows the autocorrelation function of the processed data, which still oscillates around zero apart from the zero delayed time, indicating the presence of periodic trends. It is challenging to totally eliminate the periodicity using difference or differential operators, making it difficult to obtain bit sequences that can pass the NIST SP 800-22 test.

Response to the report from the Referee #2

Comments: *“In the revised manuscript, the authors have well addressed my concerns. In my opinion, this work does not aim to promote the scientific advance of chaotic microcomb physics but made substantial efforts to apply this emerging technology for practical applications such as RNG and scalable decision making. Thus, I think the successful achievement of interesting application demonstrations of integrated chaotic microcomb sources can surely meet the criteria of Nature Communications acceptance.”*

Our reply: We thank the reviewer’s positive comments and review our work again.

Comment 1: *“I can recommend this work for publication, but I still have a minor suggestion that the authors could highlight the work (already cited as Ref [11]) on parallel LiDAR sensing based on integrated chaotic microcombs, which is in fact one of the early applications of on-chip chaotic microcomb sources.”*

Our reply: Thanks for the suggestion. We have highlighted ref [11] in the revised manuscript:

(Main part, page 2, line 14) **Especially for parallel ranging, the chaotic properties of chaotic combs were firstly harnessed for unambiguous and interference-free Lidar with simplified systems [11,36].**

Response to the report from the Referee #3

Comments: *“The replies given by the authors are satisfactory. However, I must insist that the given work is by far not the first that proposes to harness microcomb-based chaos. Therefore, I find the title, and the general presentation of the article very misleading.”*

Our reply: We thank the referee for carefully reading our manuscript and response letter. We agree with the referee that our work is not the first demonstration harnessing chaotic microcombs. As we mentioned in the original manuscript, “chaotic combs have been employed as parallel optical sources for optical coherence tomography and parallel ranging, but the parallelization was based on frequency domain rather than on the chaotic signal carried by each channel”. Thus, one of the contributions of our work is the investigation of the inter-channel correlation of chaotic combs used in this work. To making our contributions clearer, we revised the title to **“Harnessing microcomb-based parallel chaotic sources for random number generation and optical decision making”**. The Abstract is revised as following:

(Abstract, page 1, line 19) **Optical chaos is vital for various applications such as private communication, encryption, anti-interference sensing, and reinforcement learning. Chaotic microcombs have emerged as promising sources for generating massive optical chaos. However, their inter-channel correlation behavior remains elusive, limiting their potential for on-chip parallel chaotic systems with high throughput. In this study, we present massively parallel chaos based on chaotic microcombs and high-**

nonlinearity AlGaAsOI platforms. We demonstrate the feasibility of generating parallel chaotic signals with inter-channel correlation <0.04 and a high random number generation rate of 3.84 Tbps. We further show the application of our approach by demonstrating a 15-channel integrated random bit generator with a 20 Gbps channel rate using silicon photonic chips. Additionally, we achieved a scalable decision-making accelerator for up to 256-armed bandit problems. Our work opens new possibilities for chaos-based information processing systems using integrated photonics, and potentially can revolutionize the current architecture of communication, sensing and computations.

At the Introduction part, we highlighted previous works and emphasized our investigation of inter-channel correlation:

(Main part, page 2, line 11) Recently, chaotic combs have been employed as parallel optical sources for optical coherence tomography and parallel ranging. Especially for parallel ranging, the chaotic properties of chaotic combs were firstly harnessed for unambiguous and interference-free Lidar with simplified systems [11,36]. However, the parallelization in previous works was based on frequency domain rather than on the chaotic signal carried by each channel.

(Main part, page 2, line 26) In this work, we fill this gap by providing a full investigation of chaotic comb lines, and introduce the optical chaotic comb as a massively parallel chaotic source with low inter-channel correlation.

We hope these revisions making the manuscript clearer and addressing your concern.

REVIEWERS' COMMENTS

Reviewer #1 (Remarks to the Author):

I accept the manuscript in its current form.

However, I still have a concern regarding 120GS/s random rate, which would necessitate a 60GHz BW of the entropy source.

As pointed by the authors in the reply: "...But it is possible to sampling faster than the Nyquist rate with employing post-processing to obscure inter-sampling correlation. While this will not increase the rate of entropy production, which limits the random bit generation rate."

Reviewer #2 (Remarks to the Author):

The authors have addressed all my technical concerns and the revised manuscript is satisfactory—this warrants the acceptance by Nature Communications.

Reviewer #3 (Remarks to the Author):

To my point of view, the current version of the manuscript can be accepted for publication.

We appreciate the careful review by the reviewers. Here, we present a point-by-point reply (in blue) to the reviewers' comments (in black), as well as the action taken (in red).

Response to the report from the Referee #1

Comments: *"I accept the manuscript in its current form.*

However, I still have a concern regarding 120GS/s random rate, which would necessitate a 60GHz BW of the entropy source.

As pointed by the authors in the reply: "...But it is possible to sampling faster than the Nyquist rate with employing post-processing to obscure inter-sampling correlation. While this will not increase the rate of entropy production, which limits the random bit generation rate."

Our reply: Thank you for your review and for accepting the manuscript in its current form. We appreciate your positive evaluation and endorsement of the work. Your feedback and support have been invaluable throughout the review process.

As for the concern of the random number generation rate and the bandwidth, it is worth noting that the bandwidth is not the sole contributor to the entropy rate. Consider two random signals: one with a symbol rate of 10 GHz and 2 different symbols (1 bit per symbol), and the other with a symbol rate of 5 GHz and 8 different symbols (3 bits per symbol). Despite the second signal having a lower Nyquist rate than the first, the extracted random bit rate is 10 Gbps for the first signal and 15 Gbps for the second signal. Hence, to achieve a random bit generation rate of 120 Gbps, it is not necessary to produce a signal with a bandwidth of 60 GHz.

In addition, as we pointed in the previous reply: "it is possible to sample faster than the Nyquist rate with employing post-processing to obscure inter-sampling correlation." This is widely used for random bit extraction from entropy sources. As we listed in table. 1, the generation rates are faster than the twice of the bandwidth.

We hope this addresses your concern. Thank you for bringing up this point, and we appreciate your valuable feedback.

Response to the report from the Referee #2

Comments: *"The authors have addressed all my technical concerns and the revised manuscript is satisfactory—this warrants the acceptance by Nature Communications."*

Our reply: We are delighted to learn that the reviewer is satisfied with our revised manuscript and has recommended its publication. We extend our heartfelt gratitude to the reviewer for dedicating their time to reviewing our work and providing valuable comments.

Response to the report from the Referee #3

Comments: *"To my point of view, the current version of the manuscript can be accepted for publication."*

Our reply: We appreciate your perspective and are pleased to hear that you believe the manuscript is

suitable for publication. We are grateful for your time and expertise in assessing the manuscript.